# Self-supervised Learning for Incomplete Multimodal Wearable Sensor Data

## Abstract

Foundation models, a cornerstone of recent advancements in machine learning, have predominantly thrived on complete and well-structured data. However, wearable sensor data frequently suffers from significant missingness, posing a substantial challenge for the training of generalist models in this domain. This paper introduces Adaptive and Inherited Masking (`AIM`), a novel self-supervised learning (SSL) approach that learns robust representations directly from incomplete data without requiring explicit imputation. Leveraging `AIM`, we develop `AIM_FM`, a foundation model pre-trained on 40 million hours of fragmented multimodal wearable sensor data. We find that with `AIM` this model exhibits improved scaling and performance across a diverse range of tasks as compared to current state-of-the-art wearable-sensor foundation models trained on imputed data. Critically, `AIM_FM` maintains high performance even under targeted missingness scenarios (e.g., absent sensors, contiguous missingness). We will release our metabolic study dataset with reproducible training+evaluation code.

## 1 Introduction

Missingness is a natural, and often unavoidable, artifact of data in a variety of domains. Sensor systems are prone to incomplete data streams due to strategic intermittent deactivation for energy conservation, environmental noise, sensor obstruction, or hardware malfunctions (Du et al., 2020; Bähr et al., 2022; Decorte et al., 2024). Missing data is especially prevalent for mobile and wearable sensors. In addition to the aforementioned causes, user compliance issues (e.g., improper/insecure device attachment) and challenges unique to mobile devices (e.g., data transmission failures, battery charging periods) further exacerbate this problem (Rahman et al., 2017).

Self-supervised learning (SSL) has emerged as a powerful method for learning transferable representations for biosignals (Logacjov, 2024) by exploiting the inherent structure within unlabeled data (Ericsson et al., 2021). When applied at sufficient scale, these methods result in models with learned representations capable of generalizing to diverse downstream tasks, referred to as foundation models (FM) (Oquab et al., 2023; Team et al., 2023). These methods have enabled the development of wearable sensor FMs useful across a number of health prediction tasks (Narayanswamy et al., 2024a; Xu et al., 2024; Saha et al., 2025; Abbaspourazad et al., 2023).

Unfortunately, state-of-the-art (SOTA) time-series SSL approaches require fully-observed data, making it challenging to appply them directly to biosignals collected from wearables. A subset of wearable sensor FMs have therefore focused on short context windows (i.e. <60s (Abbaspourazad et al., 2023), 2.56s (Xu et al., 2024), 10s (Pillai et al., 2025)), where incomplete observations are easily filtered out. However, many critically-important physiological and behavioral patterns (e.g., circadian rhythms (Zielinski et al., 2014) and activity profiles (Hecht et al., 2009)) require the analysis of longer (hours, days) context windows, where filtering would be ineffective. This is highlighted in our data (detailed in Section 3), where 100% of the day-long samples contain some amount of missingness. While other works have used imputation (e.g. mean filling, linear interpolation) to address this challenge in developing long-context wearable FMs (Narayanswamy et al., 2024a; Erturk et al., 2025), prior work has established the difficulty of performing accurate and non-biased imputation in the face of substantial missingness (Xu et al., 2022; Shadbahr et al., 2023). Thus, there is a clear need for an approach to pre-training on time series data which is robust to data missingness.

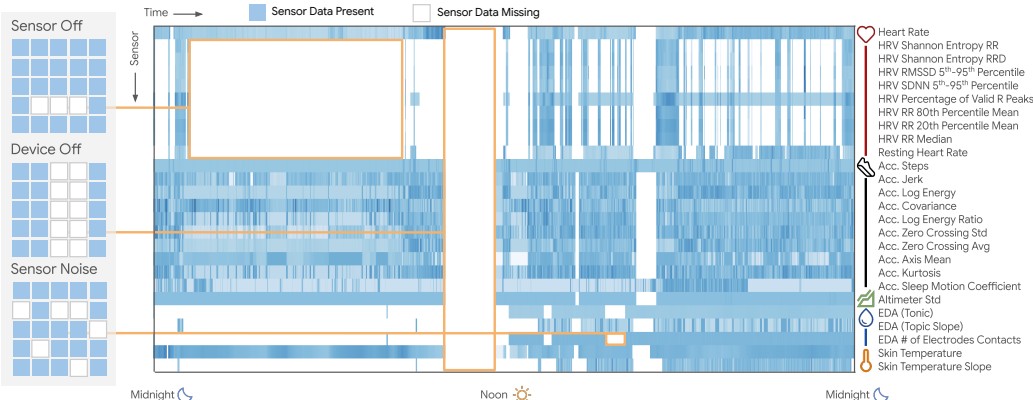

Figure 1: **Representative Example of the Sensor Data.** Our day-long wearable sensor data is composed of 26 minutely-aggregated features, derived from 5 sensors (PPG, Accel, Altimeter, EDA, and Temperature). For example, "Steps" is derived from Accel, but our data does not have the raw Accel signal itself. Please see Table 7 for exact feature definitions. Such multimodal long-context data contains complex missingness patterns (shown in white). Missingness modes include sensor(s) being off/unavailable, periods where all measures are unavailable (device is off body), and measurements that are filtered out due to being clearly spurious.

The Masked Autoencoder (MAE) SSL method learns a strong generalizable representation by introducing mask tokens to replace existing samples and then learning to reconstruct them (He et al., 2022; Narayanswamy et al., 2024a). Our key intuition is that we can co-opt this masked token to also represent the existing missingness that is naturally present in wearable sensor data. This unified treatment of missingness (natural and artificial) within a single MAE framework enables, for the first time, robust SSL without imputation, thereby avoiding any associated imputation biases that may occur (Shadbahr et al., 2023). However, the standard MAE cannot be applied with this idea because the natural missing data patterns are highly variable, and the standard MAE approach assumes that the masking ratio is fixed in order to ensure consistent batching for effecient scalability.

In this paper, we propose Adaptive and Inherited Masking, `AIM`, an SSL approach that learns representations directly from incomplete multimodal wearable sensor data with complex missingness patterns. Extending masked (MAE) pre-training (He et al., 2022), `AIM` is able to flexibly handle variable mask tokens while retaining the computational advantages of the original MAE framework that allows it to conduct large-scale pre-training. The learnable mask token is shared to represent two different types of missingness: *Inherited* missingness, which is a mask inherited from the natural missingness, and *Artificial* missingness, which is a mask applied on observed data to formulate the masked-reconstruction SSL task. Futhermore, `AIM`'s flexibility of variable missingness enables the use of a diverse mix of *artificial* masking strategies that, for the first-time, utilize strategy-specific masking ratios, mimicking real-world structured failure modes common to wearable sensor data.

**The key contributions of our work are:**

1. We propose `AIM`, a simple yet powerful method for large-scale pre-training on data with missingness. `AIM` jointly models *inherited* (real-world) masking with a diverse mix of *artificial* masking strategies with strategy-specific ratios to learn the complex missingness patterns in wearable sensor streams.

2. After pre-training on 40m hours of fragmented wearable data, `AIM_FM` is a foundation model that exhibits improved scaling and downstream performance on a diverse range of task semantics (cardiovascular, mental health, motion, demographics, metabolics).

3. We benchmark against general SSL methods, an irregularly-sampled TS method, a forecasting foundation model, and five wearable-specific methods that had been trained on imputed data. In so doing, *we demonstrate that the standard practice of imputating missing data during pre-processing, used by SOTA FMs, is not only unnecessary, but is actually suboptimal*.

4. We evaluate the robustness of `AIM_FM` across a wide range of targeted missing scenarios, dropping out specific sensors or time windows, and we demonstrate 73% less average performance degradation as compared to baselines pre-trained with imputed data.

5. We will release our full metabolic study dataset (used for anxiety, hypertension, insulin resistance, age, and BMI tasks), the `AIM_FM` model weights trained on this data, and a codebase with the full reproducible evaluation code upon acceptance. See Reproducibility Statement for details.

## 2 RELATED WORK

**Our Long-context Setting vs. Other Short-Context Biosensor Settings**. While we also focus on wearable biosensor data, our setting differs significantly from traditional settings. For example, instead of high-frequency sensor data, our data is composed minutely aggregated features derived from a multimodal sensor suite (e.g. Step Count from IMU). Please see Fig. 1 for a data example, and Table 7 for exact feature definitions. As such, we focus on learning a representation for general physiological and behaviorial sensing from day-long data, rather than archetypal seconds-long wearable models for just-in-time interventions and detection (Vandelanotte et al., 2023).

Many wearable methods make assumptions on the data, which do not hold true in our setting. For example, recent HAR methods, CrossHAR (Hong et al., 2024) and UniMTS (Zhang et al., 2024) use physical 3D rotations during training, but our data does not have specific x,y,z axes. Furthermore, there are no public datasets that match our setting. A prominent HAR dataset, PAMAP2 (Reiss & Stricker, 2012) only utilizes IMU sensors with only 10 seconds of data per label. More broadly, WESAD (Schmidt et al., 2018) is a wearable dataset that contains a variety of sensing modalities (i.e. PPG, Accel, EDA, and temperature), but only has 1 minute of data per label. All of Us (Jeong et al., 2025) does have real-world day-long sensor data, but is limited to only the 2 channels, compared to our 26. To help address this, we will release our metabolic study dataset.

Table 1: **Comparison of Settings.** We use Human Activity Recognition (HAR) as a comparison point as the representative of the traditional biosensor domain. Activity is included as one of our many eval tasks, but it is defined very differently from the traditional HAR domain. In our setting, an activity is "retroactively" predicted from an entire day's worth of data. In the traditional setting, an activity is "just-in-time" predicted from a few seconds.

| | Resolution | Context | Modalities | Eval Tasks |
|---|---|---|---|---|
| Our Setting | Minutely (0.02 Hz) | Day-long | IMU, PPG, EDA, Temp, Altimeter | Physiology (HT, IR) Mental (Anxiety) Motion (Activity) Demographic (Age, BMI) |
| HAR Setting | High Freq (~100 Hz) | Seconds-long | IMU | Motion (Activity, Gait) |

**Self-Supervised Learning for Time-Series Foundation Models.** Time-series FMs typically leverage one of two classes of SSL pre-training. The first body of work applies a constrastive objective where data pairs are typically generated via augmentations (Tang et al., 2020), temporal proximity (Tonekaboni et al., 2021), subject labels (Abbaspourazad et al., 2023), domain knowledge (Pillai et al., 2025), or motif similarity (Xu et al., 2023; 2024). While powerful, these methods rely on strong assumptions to currate pairs. A second line of work implements a generative objective, often masked reconstruction (He et al., 2022). These works typically focus exclusively on complete univariate signals (Dong et al., 2023; Li et al., 2023; Chien et al., 2022), model highly correlated channels from a single modality (Na et al., 2024), or focus on task-specific forecasting without learning transferrable embeddings (e.g. Chronos (Ansari et al., 2024), TimesFM (Das et al., 2024)). The recent past has seen large-scale SSL extended to long-context multi-modal wearable sensor data (Narayanswamy et al., 2024a; Erturk et al., 2025). However, these SOTA wearable FMs opt to use naive imputation to handle their ubiquituous missingness. In contrast, `AIM_FM` leverages masked reconstruction pre-training to jointly model existing missingness.

**Self-supervised Learning for Other Incomplete Data.** SSL methods for other incomplete data have typically focused on tabular inputs with simple, point-wise missingness (Ucar et al., 2021; Chang et al.) or irregularly-sampled time-series (Beebe-Wang et al., 2023). Irregularly-sampled time-series represents a similar but different domain from our missingness-afflicted mobile health setting. Irregularly sampled time-series such as ICU lab measurements (Silva et al., 2012) collected at distinct intervals with many other modalities typically missing, whereas wearables produce regularly-sampled data where modalities will drop out in structured groups (Figure 2). The most related work is mTAN (Shukla & Marlin, 2021), which proposes an approach and architecture for learning self-supervised representations directly on multimodal ICU data without imputation. Neural ODEs also learn directly on the irregularly-sampled time-series (Rubanova et al., 2019), but have high computational cost and do not necessarily scale to our large-scale pre-training setting (Finlay et al., 2020). We benchmark `AIM_FM` against mTAN and show that our model achieves better performance across every task.

**Supervised Learning for Incomplete Data.** A majority of these works focus on supervised imputation, and there are many on multivariate time-series imputation (Yoon et al., 2018; Qin & Wang, 2023; Dai et al., 2024). A few works have investigated how modeling existing missingness can help aid in improving imputation accuracy (Du et al., 2023; Wei et al., 2024). For supervised classification,

a handful of works have explored how imputation can introduce bias in classifiers trained on the imputed data (Jungo et al., 2024; Shadbahr et al., 2023; Xu et al., 2022), and a few have proposed methodologies for learning supervised classifiers directly on the missing data (Ghahramani & Jordan, 1993; Ipsen et al., 2022). Recently, a few works have focused on the "missing modality" setting, where a modality will be missing across the entire time-series (Nie et al., 2023; Han et al., 2024). Our setting is a generalization of this, where specific modalities will drop in and out over time due to a variety of factors, such as packet loss, strategic de-activation for battery conservation, and intermittent loosening. Please see Fig. 1 for a visualization of our missingness modes. Also, it is worth noting that it is unclear how to extend these supervised methods for self-supervised learning. AIM is able to unify its approach learn a representation directly from the incomplete data, demonstrating a simple yet powerful approach for large-scale pre-training.

## 3 Large Scale Incomplete Wearable Data

A primary contribution of our work is the modeling of incomplete data during pre-training and inference. We curate a large pre-training dataset in addition to two labeled datasets for downstream tasks. Each data sample is comprised of 26 features derived from 5 sensors (photoplethysmography, accelerometer, skin conductance, altimeter, and temperature) and sampled once per minute for a duration of 1440 minutes (1 day). An 80/20 train/test split among the participants was used for each dataset and designed to not overlap across pre-training and downstream datasets. A core artibute of wearable sensor data is its complex and often structured missingness patterns. A representative example of sensor data missingness is illustrated in Fig. 1. We note that pre-training and downstream data are derived from similar devices and thus exhibit similar missingness patterns. We further note that missingness is ubiquitous in long-context sensor data, with *0% of the samples over our dataset (1.6 million day-long windows) exhibiting 0% missingness.* Please refer to the Appendix A.1 for further data descriptions.

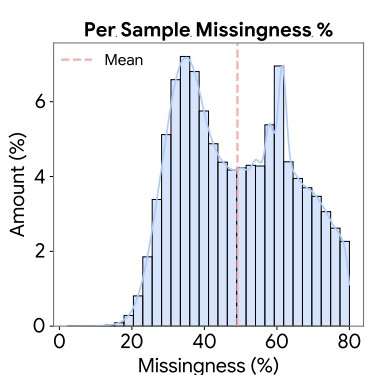

Figure 2: **Distribution of Missingness % Per Sample.** Mean 49%, Minimum 2%, Maximum 80%.

**Pre-training Data.** For pre-training, we used a de-identified dataset collected between [3/1/2024-6/1/2024]. The dataset included 1,601,088 instances of day-long data with 40 million total hours. This data originates from 27,137 unique individuals, with a mean of 59 days contributed per participant.

**Downstream Activity Study Data.** This data originates from the same source as our pre-training data. We randomly sampled up to 5,000 examples for each of 20 activities for training and up to 1,000 examples of each activity for testing. These self-reported activities span common exercises like walking, gym-based training like weight lifting, and sports like skiing. In total, 104,086 activities were sampled from 46,199 people. The mean duration per activity was 66 minutes.

**Downstream Metabolic Study Data.** This data originates from an IRB approved observational study, in which participants consented to data sharing. In total, the data comprises 5.8M person-hours of wearable sensor data (241,532 day-long instances), collected from 1,250 individuals. Downstream targets include self-reported medical conditions (hypertension, anxiety) and demographics (age, BMI), as well as insulin resistance measurements, which were calculated from fasting insulin and glucose lab tests. *Upon acceptance, we will release this data, `AIM_FM` model weights trained on this data, and a codebase for reproducible evaluation of downstream targets.* This release will provide a valuable community resource by expanding the data available for wearable foundation model training by several orders of magnitude and providing a unified benchmarking task.

## 4 AIM Methodology

AIM is a **simple yet powerful idea** that allows us to learn a missingness-aware foundation model. The specific methodological contributions are as follows:

- First to jointly model inherited and artificial missingness for representation learning, enabling the model to recognize missingness as a natural feature of the data.

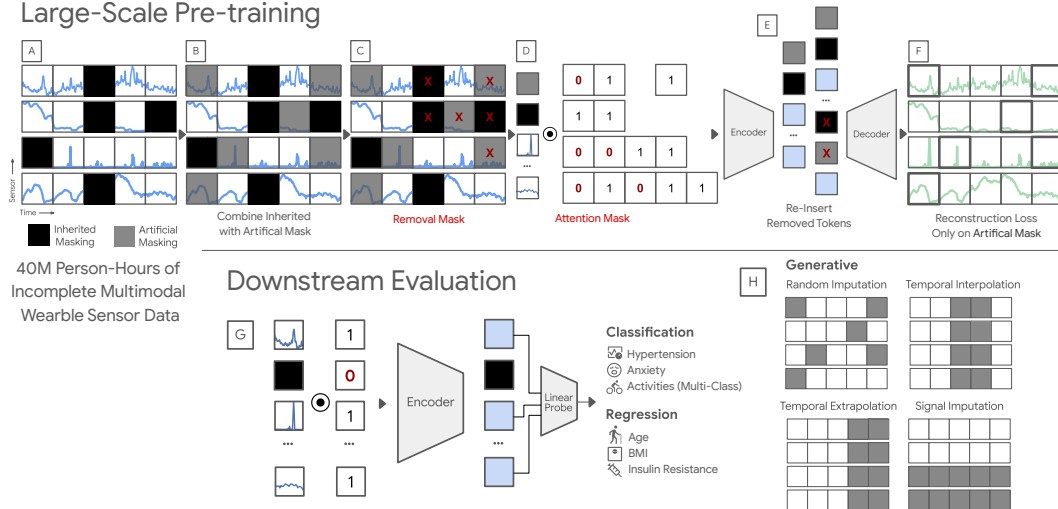

Figure 3: **AIM Pre-training [A-F] and Evaluation [G,H] Methodology**. Our mask is the union of **[A]** inherited missingness from real-world noise and **[B]** artificial masking of observed data. Both are via a shared learnable mask token. Because the inherited mask introduces variable masking, **[C]** we first remove $D$ (size of artificial mask) tokens and **[D]** then use an attention mask to remove the remaining. **[E]** Dropped tokens are reinserted before **[F]** the final reconstruction. **[G]** Reconstruction error is computed only on artificial masks with known ground truth. **[H]** For discriminative tasks, a linear probe is trained on a pooled representation of the non-missing data.

- First to adapt MAE to allow for variable masking ratios without sacrificing computational efficiency for large-scale pre-training.
- First to propose training with a concurrent mix of masking strategies (i.e. random, temporal bar, modality bar), rather than a single masking strategy (i.e. random masking).

Before the MAE method was published, masked reconstruction was already a widely-studied method in machine learning. MAE established the key details that made it a competitive pre-training method. Analogously, before AIM, there was no clear understanding of what constituted an effective pre-training strategy for data in the presence of missingness. While recent baselines (Erturk et al., 2025; Narayanswamy et al., 2024a) continue to train on imputed data, AIM demonstrates that the standard imputation approach is not only unnecessary, but is actually suboptimal.

**Motivation.** While missingness is ubiquitous in wearable sensor data, SOTA FMs fail to gracefully handle this and instead opt to naively apply simple imputation methods (Narayanswamy et al., 2024a; Erturk et al., 2025), which can potentially bias the model (Shadbahr et al., 2023). Our key insight is to inherit these pre-existing missingness patterns to be used in conjunction within an masked pre-training framework (He et al., 2022). By treating *inherited* missingness as a natural artifact of sensor data, equal to the *artifical* masks used as reconstruction targets, AIM establishes missingness as an inherent structure embedded in the learned representation during pre-training.

AIM first takes an input matrix of sensor features, which are then tokenized to be $\mathbf{X} \in \mathbb{R}^{B \times N \times E}$ ($B$ is batch size, $N$ is number of tokens, and $E$ is embedding dimension). We then define a binary vector mask, $\mathbf{M} \in \{0, 1\}^{B \times N}$ (where 1 is masked and 0 is non-masked) equal in length to the number of tokenized sensor inputs, where masked tokens are ignored by the encoder. Our method sets $\mathbf{M}$ as the union of the *inherited* and *artificial* masks such that: $\mathbf{M} = \mathbf{M}^{\text{inherited}} \vee \mathbf{M}^{\text{artificial}}$. The *inherited* mask, $\mathbf{M}^{\text{inherited}}$, represents inherent missingness. The *artificial mask*, $\mathbf{M}^{\text{artificial}}$, is simulated missingness on observed data used in the reconstruction training objective. Critically, the inclusion of the inherited mask ensures that the encoder exclusively learns representations from reliable, observed, sensor data without contamination from imputation artifacts.

**Background.** The original MAE work (He et al., 2022) implements masking through the *removal mask*, where masked tokens are removed from the token sequence processed by the encoder. By dropping $D$ tokens per sample, the *removal mask* reduces the computation of the transformer encoder from $O(N^2) \rightarrow O((N-D)^2)$ (25x less compute when masking 80% of tokens). Though computationally efficient, the *removal mask* generally requires a fixed value $D$ such that $\sum_{n=1}^{N} \mathbf{M}_{[b,n]} = D \ \forall \ b \in [1, B]$. This is to ensure that the masked input $\mathbf{X}[\mathbf{M}] \in \mathbb{R}^{B \times (N-D) \times E}$ fed to the transformer encoder is of a fixed size. Consequently, prior MAE-based SSL methods have

Table 2: **Pre-training Masking % Sweep.** Each table shows the effect of varying a given pre-training strategy's mask % on its generative evaluation counterpart. The gray row highlights the best pre-training ratio. The best results balance consistent performance across eval ratios and prefer higher pre-training % when results are similar, in order allow for better effeciency with a higher removal $D$. Thus, our pre-training masking mix is 80% random, 50% temporal slice, and 50% sensor slice.

(a) *Random Imp Pre-train*

| PT Mask % | Eval Ratio | | |
|---|---|---|---|
| | 30% | 50% | 80% |
| 90% | 0.13 | 0.14 | 0.20 |
| **80%** | 0.10 | 0.12 | 0.19 |
| 70% | 0.10 | 0.12 | 0.19 |
| 60% | 0.10 | 0.12 | 0.19 |
| 50% | 0.09 | 0.12 | 0.20 |

(b) *Temporal Slice Pre-train*

| PT Slice % | Eval Amount | | | |
|---|---|---|---|---|
| | 10m | 30m | 60m | 180m |
| 70% | 0.23 | 0.34 | 0.41 | 0.56 |
| 60% | 0.26 | 0.36 | 0.42 | 0.57 |
| **50%** | 0.23 | 0.33 | 0.40 | 0.55 |
| 40% | 0.22 | 0.33 | 0.40 | 0.56 |
| 30% | 0.22 | 0.33 | 0.40 | 0.57 |

(c) *Signal Slice Pre-train*

| PT Slice % | Eval Amount | | | |
|---|---|---|---|---|
| | 2/26 | 6/26 | 12/26 | 24/26 |
| 70% | 0.19 | 0.23 | 0.28 | 0.43 |
| 60% | 0.18 | 0.22 | 0.27 | 0.45 |
| **50%** | 0.17 | 0.21 | 0.27 | 0.48 |
| 40% | 0.17 | 0.21 | 0.27 | 0.56 |
| 30% | 0.16 | 0.21 | 0.30 | 0.63 |

Metrics: Mean Squared Error

traditionally required fixed masking ratios (He et al., 2022; Narayanswamy et al., 2024a; Girdhar et al., 2023; Huang et al., 2022; Tong et al., 2022).

Unfortunately, modeling sensor missingness via fixed removal amount poses a significant challenge as missingness is naturally variable. This can be addressed by passing all tokens to the encoder and using an *attention mask* instead. An *attention mask* method would use the transformer's innate attention mechanism, setting the attention weights for masked tokens to zero, preventing them from contributing to the encoder output (Vaswani et al., 2017; Du et al., 2023). While flexible, passing all tokens through the encoder is computationally prohibitive for long sequences and large scale pre-training.

**Taking AIM with Adaptive Inherited Masking.** The key insight of AIM is to unify the efficiency of the removal mask with the flexibility of attention masking. This hybrid strategy allows for the handling of data with variable, inherited missingness while retaining the computational advantages of the original MAE framework. The process, visualized in Fig. 3, operates as a two-stage approach to handle the total set of masked tokens (which includes *inherited* and *artificial* masked tokens). First, to guarantee efficient computation, $D$ tokens, a subset of all masked tokens, are *removed* from the sequence fed to the encoder. $D$ is determined as the lower bound of possible masked tokens, and can be set to the artificial mask ratio during training. Second, remaining masked tokens, not previously *removed*, are masked via the encoder's *attention mechanism*. Specifically, for this variable number of tokens, the attention weights are set to 0, preventing these tokens from contributing to the encoder's internal representation. In this way AIM extends masked pretraining to support variable *inherited* and *artificial* missingness while retaining the computational benefits of MAE needed for scalable pre-training.

**AIM Enables Complex Masking Mixtures.** As previously discussed, MAE-based methods have traditionally been constrained to fixed masked ratios, and by convention have used only one fixed masking strategy (He et al., 2022; Narayanswamy et al., 2024a; Girdhar et al., 2023; Huang et al., 2022; Tong et al., 2022). AIM eliminated this requirement by efficiently handling variable masking through a combination of removal and attention masking, enabling a novel heterogenous mixture of *artificial* masking strategies and ratios, simulating the complex modes of data loss seen in real-world sensor streams (Fig. 1). For instance, as determined in Table 2, random masking benefits from a high 80% ratio, as tokens are easily reconstructed from neighbors, while the more challenging slice objectives benefits from a lower 50% ratio. Specifically, during training, each input window randomly uses one of the following three distinct masking strategies to model domain specific missingness patterns:

1. *Random Imputation Pre-training:* Drops a percentage of total tokens in a point-wise fashion to simulate sensor noise where individual channels fail at random times.
2. *Temporal Slice Pre-training:* Drops all sensor channel data for a percentage of total time slices. This models "off body" events, where a wearable is temporarily removed.
3. *Sensor Slice Pre-training:* Drops a percentage of sensor channels entirely across all time points. This simulates "sensor off" events, for instance, to conserve battery life.

**AIM is a Unified Framework for Pre-training and Evaluation.** AIM provides a unified framework that consistently handles missing data during both pre-training and evaluation. The full pre-training procedure can be seen in Fig. 3 [A-G]. AIM does not differentiate between *inherited* or *artificially* masked tokens, encouraging the model to understand fragmentation as an innate aspect of multimodal sensor data. Crucially, AIM's adaptive masking can also be leveraged during evaluation, as illustrated in Fig. 3 [G,H]. The AIM pre-trained model is able to operate directly on incomplete multimodal sensor data by dynamically attending only to observed segments. This eliminates the need to impute or discard missing values, and thus ensures generalization from pre-training to downstream inference.

## 5 Experiments

Here, we describe our experimental design. See Appendix A.3 for additional implementation details.

**Pre-training.** We pre-train `AIM_FM` on minutely multimodal wearable data ($\mathbf{A} \in \mathbb{R}^{N \times T \times S}$) where $S = 26$ sensor features, $T = 1440$ minutes, and $N = 1,601,088$ is the total dataset size. Each signal modality is tokenized with a shared 1D convolution with a kernel size and stride of 10 minutes. The tokenized output is of size 144 x 26, for 3,744 total tokens. A 2D sinusoidal positional embedding is used to encode time and signal identity and is added to the token representations before being passed to the ViT-1D encoder/decoder. `AIM_FM` has 25M parameters, 384-d hidden size, 12 encoder layers, and 4 decoder layers. Following Section 4, we apply a composite mask (80% random, 50% temporal, 50% signal slices) and optimize mean squared error over *artificially* masked patch reconstruction. Training is performed on 8x16 Google v5e TPUs with a batch size of 512 for 100K steps.

**Baselines.** SSL baselines are trained from scratch using the same pre-training set-up, unless otherwise noted. They include SOTA baselines from the wearable space, as well as common self-supervised learning methods. Crucially, *all baselines use imputed data* to meet their complete-input requirement.
- *LSM* (Narayanswamy et al., 2024a): A SOTA wearable FM leveraging a vanilla MAE framework with a ViT-2D backbone. It trains on imputed data and relies on a fixed masking stategy and ratio.
- *WBM-TST* (Erturk et al., 2025): A SOTA wearable FM trained on low-frequency, multimodal wearable sensing data with a ViT-1D backbone. It pre-trains with subject-aware contrastive learning.
- *LIMU-BERT* (Xu et al., 2021): An SSL method developed for wearable data. It uses an reconstruction objective that masks across all signals for given time points.
- *RelCon* (Xu et al., 2024): A SOTA wearable FM method for high-frequency, uni-modal data.
- *mTAN* (Shukla & Marlin, 2021): An method and archecture for irregularly sampled time-series.
- *TimesFM 2.0* (Das et al., 2024): A SOTA forecasting foundation model with 500m parameters trained across a variety of domains for zero-shot temporal extrapolation.
- *SimCLR* (Chen et al., 2020) / *DINO* (Caron et al., 2021) / *MSN* (Assran et al., 2022): General contrastive learning SSL methods with empirically-validated temporal augmentations (Liu et al., 2024).

**Downstream Evaluation.** We evaluate `AIM_FM` across three downstream targets: generative, classification, and regression. For `generative`, we assess reconstruction under structured missingness patterns: (1) random imputation (30%, 50%, 80%), (2) temporal interpolation (contiguous masked windows of 10, 30, or 60 minutes), (3) temporal extrapolation (masked window at the end of the sequence), and (4) signal imputation (masking 2/26, 6/26, or 12/26 channels). Since contrastive baselines lack reconstruction objectives, we compare against LSM (Narayanswamy et al., 2024a) in addition to simple imputation methods used in practice—Linear Interpolation, Nearest Neighbors, and Mean Filling—under the same union masking scheme. We omit MICE (Van Buuren & Groothuis-Oudshoorn, 2011) as its missingness-at-random assumptions do not hold and its poor performance in prior work (Narayanswamy et al., 2024a). For `classification`, we average embeddings over non-inherited-masked tokens and apply a trainable linear probe; LSM pools across all tokens, and contrastive methods use the CLS token. We report $F_1$, Accuracy, Balanced Accuracy, and AUROC on targets including hypertension, anxiety (Metabolics dataset; see Section 3), and 20-class activity recognition (Activity dataset). For `regression`, we follow the same setup with a linear regression probe and report MAE and Pearson correlation on BMI, age, and insulin resistance (Metabolics dataset). Confidence intervals were calculated via 100 bootstrap iterations.

## 6 Results and Discussion

**Generalizability Across Generative, Classification, and Regression.** `AIM_FM` learns a generalizable representation, useful for generative, classification, and regression tasks (Tables 3, 4, 5).

`AIM_FM` demonstrates a dramatic 35.6% average gain across the 12 generative tasks compared against LSM, the most closely related work, implying that training on imputed data may negatively bias LSM's generative capabilities. While both methods use random imputation during pre-training, our method's modeling of *inherited* missingness enables an average improvement of 21% for random imputation. In addition to the random masking strategy, `AIM`'s ability to mix masking strategies (such as signal/temporal slice masking) enable strong performance on more strucutured generative tasks such as temporal interpolation/extrapolation and signal imputation. `AIM_FM` is even able to achieve stronger performance than the forecasting-specific time-series foundation models, TimesFM,

Table 3: **Generative Task Results**

| | | ↓Random Imp. | | | ↓Temporal Interp. | | | ↓Temporal Extrap. | | | ↓Signal Imp. | |
|---|---|---|---|---|---|---|---|---|---|---|---|---|---|
| | Method | 30% | 50% | 80% | 10m | 30m | 60m | 10m | 30m | 60m | 2 | 6 | 12 |
| Stats | Linear Int. | $0.572_{\pm.003}$ | $0.629_{\pm.004}$ | $0.788_{\pm.006}$ | $0.523_{\pm.012}$ | $0.713_{\pm.013}$ | $0.850_{\pm.013}$ | $0.749_{\pm.021}$ | $0.983_{\pm.023}$ | $1.174_{\pm.026}$ | - | - | - |
| | NN Fill | $0.707_{\pm.003}$ | $0.773_{\pm.004}$ | $0.952_{\pm.006}$ | $0.648_{\pm.015}$ | $0.868_{\pm.014}$ | $1.030_{\pm.014}$ | $0.749_{\pm.020}$ | $0.983_{\pm.023}$ | $1.174_{\pm.026}$ | - | - | - |
| | Mean Fill | $0.924_{\pm.005}$ | $0.957_{\pm.005}$ | $0.949_{\pm.006}$ | $0.929_{\pm.015}$ | $0.932_{\pm.016}$ | $0.963_{\pm.014}$ | $1.096_{\pm.022}$ | $1.096_{\pm.018}$ | $1.086_{\pm.016}$ | $1.238_{\pm.025}$ | $1.258_{\pm.015}$ | $1.268_{\pm.015}$ |
| DL | TimesFM | - | - | - | - | - | - | $0.548_{\pm.051}$ | $0.687_{\pm.060}$ | $0.857_{\pm.059}$ | - | - | - |
| | Limu-bert | - | - | - | $1.010_{\pm.017}$ | $1.062_{\pm.016}$ | $1.062_{\pm.016}$ | $1.154_{\pm.021}$ | $1.159_{\pm.020}$ | $1.159_{\pm.015}$ | - | - | - |
| | mTAN | $0.605_{\pm.002}$ | $0.617_{\pm0.002}$ | $0.627_{\pm.003}$ | $0.647_{\pm.016}$ | $0.697_{\pm.011}$ | $0.731_{\pm.010}$ | $0.721_{\pm.014}$ | $0.803_{\pm.019}$ | $0.929_{\pm.022}$ | $0.741_{\pm.011}$ | $0.790_{\pm.007}$ | $0.958_{\pm.010}$ |
| | LSM | $0.146_{\pm.000}$ | $0.178_{\pm.001}$ | $0.293_{\pm.001}$ | $0.605_{\pm.010}$ | $0.687_{\pm.008}$ | $0.717_{\pm.011}$ | $0.670_{\pm.020}$ | $0.746_{\pm.012}$ | $0.775_{\pm.009}$ | $0.683_{\pm.041}$ | $0.561_{\pm.016}$ | $0.443_{\pm.005}$ |
| | OURS | $\mathbf{0.113}_{\pm.000}$ | $\mathbf{0.133}_{\pm.001}$ | $\mathbf{0.218}_{\pm.002}$ | $\mathbf{0.330}_{\pm.006}$ | $\mathbf{0.466}_{\pm.007}$ | $\mathbf{0.545}_{\pm.004}$ | $\mathbf{0.447}_{\pm.010}$ | $\mathbf{0.577}_{\pm.014}$ | $\mathbf{0.687}_{\pm.009}$ | $\mathbf{0.179}_{\pm.011}$ | $\mathbf{0.205}_{\pm.006}$ | $\mathbf{0.257}_{\pm.006}$ |

Metrics: Mean Squared Error | Tasks: Random Imputation (30%, 50%, 80% missing), Temporal Interpolation/Extrapolation (10, 30, 60 missing minutes), Signal Imputation (2, 6, or 12 out of 26 missing modalities) | Methods: Statistical (Top), Deep Learning (Bottom)

Table 4: **Classification Task Results**

| | | Hypertension (2) | | | | Anxiety (2) | | | | Activity Recognition (20) | | | |
|---|---|---|---|---|---|---|---|---|---|---|---|---|---|
| | Method | ↑$F_1$ | ↑Acc | ↑BAcc | ↑AUC | ↑$F_1$ | ↑Acc | ↑BAcc | ↑AUC | ↑$F_1$ | ↑Acc | ↑BAcc | ↑AUC |
| Full-Sup | ResNet | $.529_{\pm.003}$ | $.587_{\pm.003}$ | $.516_{\pm.003}$ | $.624_{\pm.004}$ | $.655_{\pm.003}$ | $.651_{\pm.003}$ | $.645_{\pm.003}$ | $.709_{\pm.003}$ | $.721_{\pm.007}$ | $.734_{\pm.007}$ | $.729_{\pm.007}$ | $.965_{\pm.002}$ |
| | ViT-1D | $.516_{\pm.003}$ | $.509_{\pm.004}$ | $.481_{\pm.003}$ | $.520_{\pm.005}$ | $.597_{\pm.004}$ | $.586_{\pm.004}$ | $.583_{\pm.004}$ | $.620_{\pm.004}$ | $.367_{\pm.008}$ | $.374_{\pm.007}$ | $.351_{\pm.008}$ | $.863_{\pm.004}$ |
| | OURS (FT) | $.680_{\pm.003}$ | $.648_{\pm.004}$ | $.626_{\pm.003}$ | $.703_{\pm.004}$ | $\mathbf{.693}_{\pm.003}$ | $\mathbf{.693}_{\pm.003}$ | $\mathbf{.685}_{\pm.003}$ | $\mathbf{.759}_{\pm.003}$ | $\mathbf{.767}_{\pm.007}$ | $\mathbf{.779}_{\pm.007}$ | $\mathbf{.774}_{\pm.007}$ | $\mathbf{.978}_{\pm.001}$ |
| Frozen + LP | Limu-bert | $.599_{\pm.003}$ | $.596_{\pm.004}$ | $.561_{\pm.004}$ | $.635_{\pm.004}$ | $.640_{\pm.003}$ | $.641_{\pm.003}$ | $.632_{\pm.003}$ | $.693_{\pm.004}$ | $.190_{\pm.006}$ | $.219_{\pm.008}$ | $.191_{\pm.006}$ | $.735_{\pm.004}$ |
| | WBM | $.582_{\pm.004}$ | $.572_{\pm.004}$ | $.542_{\pm.004}$ | $.599_{\pm.004}$ | $.605_{\pm.003}$ | $.604_{\pm.003}$ | $.597_{\pm.003}$ | $.643_{\pm.004}$ | $.107_{\pm.005}$ | $.117_{\pm.006}$ | $.102_{\pm.005}$ | $.611_{\pm.006}$ |
| | RelCon | $.564_{\pm.004}$ | $.565_{\pm.005}$ | $.530_{\pm.005}$ | $.590_{\pm.006}$ | $.615_{\pm.004}$ | $.609_{\pm.004}$ | $.604_{\pm.004}$ | $.652_{\pm.005}$ | $.058_{\pm.004}$ | $.050_{\pm.000}$ | $.005_{\pm.000}$ | $.509_{\pm.005}$ |
| | mTAN | $.637_{\pm.003}$ | $.576_{\pm.004}$ | $.566_{\pm.003}$ | $.607_{\pm.005}$ | $.506_{\pm.003}$ | $.582_{\pm.003}$ | $.486_{\pm.003}$ | $.674_{\pm.003}$ | $.193_{\pm.006}$ | $.217_{\pm.007}$ | $.192_{\pm.006}$ | $.731_{\pm.004}$ |
| | SimCLR | $.524_{\pm.003}$ | $.548_{\pm.004}$ | $.501_{\pm.004}$ | $.568_{\pm.004}$ | $.603_{\pm.003}$ | $.601_{\pm.004}$ | $.594_{\pm.004}$ | $.636_{\pm.004}$ | $.109_{\pm.003}$ | $.124_{\pm.007}$ | $.098_{\pm.006}$ | $.652_{\pm.005}$ |
| | DINO | $.536_{\pm.004}$ | $.504_{\pm.004}$ | $.487_{\pm.004}$ | $.510_{\pm.004}$ | $.557_{\pm.004}$ | $.562_{\pm.003}$ | $.551_{\pm.004}$ | $.582_{\pm.004}$ | $.110_{\pm.005}$ | $.124_{\pm.007}$ | $.102_{\pm.005}$ | $.635_{\pm.005}$ |
| | MSN | $.555_{\pm.003}$ | $.552_{\pm.004}$ | $.519_{\pm.003}$ | $.575_{\pm.004}$ | $.547_{\pm.004}$ | $.551_{\pm.004}$ | $.515_{\pm.003}$ | $.571_{\pm.004}$ | $.144_{\pm.006}$ | $.159_{\pm.008}$ | $.136_{\pm.006}$ | $.692_{\pm.005}$ |
| | LSM | $.676_{\pm.003}$ | $.682_{\pm.004}$ | $.640_{\pm.003}$ | $.739_{\pm.004}$ | $.678_{\pm.003}$ | $.678_{\pm.003}$ | $.670_{\pm.003}$ | $.743_{\pm.004}$ | $.470_{\pm.006}$ | $.489_{\pm.008}$ | $.470_{\pm.008}$ | $\mathbf{.900}_{\pm.003}$ |
| | OURS | $\mathbf{.687}_{\pm.003}$ | $\mathbf{.693}_{\pm.004}$ | $\mathbf{.651}_{\pm.003}$ | $\mathbf{.754}_{\pm.004}$ | $.690_{\pm.003}$ | $692_{\pm.004}$ | $.683_{\pm.004}$ | $.758_{\pm.004}$ | $\mathbf{.472}_{\pm.006}$ | $\mathbf{.493}_{\pm.008}$ | $\mathbf{.474}_{\pm.008}$ | $.899_{\pm.003}$ |

Metrics: $F_1$ Score, Accuracy, Balanced Accuracy, AUROC with Macro One-vs-Rest | Tasks: 20-class Activity Recognition, rest are binary | Methods: Fully Supervised with Ours Fine-Tuned (Top), SSL Methods Frozen with Linear Probe (Bottom).

which has 20x the learnable parameters. This demonstrates that `AIM` has a superior capacity to model the underlying data distribution.

Crucially, these generative gains do not compromise performance on the discriminative tasks. `AIM_FM` consistently matches or surpasses LSM, achieving the strongest performance on 14/18 classification+regression metrics across five highly diverse domains: cardiovascular, mental health, motion, metabolics, and demographics. With a simple linear probe and frozen features, our model surpasses fully supervised baselines on all tasks but activity. The 95% confidence intervals confirm that these gains are statistically significant,

Table 5: **Regression Task Results**

| | | Age | | BMI | | Insulin Resis. | |
|---|---|---|---|---|---|---|---|
| | Method | ↓MAE | ↑Corr | ↓MAE | ↑Corr | ↓MAE | ↑Corr |
| Full-Sup | ResNet | $7.429_{\pm.039}$ | $.618_{\pm.004}$ | $5.067_{\pm.028}$ | $.515_{\pm.005}$ | $1.640_{\pm.018}$ | $.241_{\pm.011}$ |
| | ViT-1D | $9.653_{\pm.049}$ | $.132_{\pm.006}$ | $6.061_{\pm.035}$ | $.047_{\pm.006}$ | $1.580_{\pm.016}$ | $.139_{\pm.009}$ |
| | OURS (FT) | $7.574_{\pm.037}$ | $.606_{\pm.004}$ | $5.172_{\pm.032}$ | $.522_{\pm.007}$ | $\mathbf{1.435}_{\pm.020}$ | $.291_{\pm.009}$ |
| Frozen + LP | Limu-bert | $8.445_{\pm.038}$ | $.475_{\pm.005}$ | $5.486_{\pm.029}$ | $.408_{\pm.004}$ | $1.599_{\pm.017}$ | $.223_{\pm.009}$ |
| | WBM | $8.614_{\pm.036}$ | $.449_{\pm.005}$ | $5.662_{\pm.029}$ | $.352_{\pm.006}$ | $1.714_{\pm.017}$ | $.227_{\pm.010}$ |
| | RelCon | $8.886_{\pm.044}$ | $.380_{\pm.006}$ | $5.527_{\pm.031}$ | $.407_{\pm.007}$ | $1.611_{\pm.017}$ | $.189_{\pm.009}$ |
| | mTAN | $9.088_{\pm.046}$ | $.334_{\pm.005}$ | $5.835_{\pm.035}$ | $.240_{\pm.007}$ | $1.577_{\pm.017}$ | $.204_{\pm.010}$ |
| | SimCLR | $9.207_{\pm.042}$ | $.345_{\pm.005}$ | $5.852_{\pm.028}$ | $.235_{\pm.005}$ | $\mathbf{1.546}_{\pm.016}$ | $.248_{\pm.009}$ |
| | DINO | $9.685_{\pm.044}$ | $.112_{\pm.006}$ | $5.968_{\pm.036}$ | $.122_{\pm.006}$ | $1.588_{\pm.016}$ | $.087_{\pm.011}$ |
| | MSN | $9.416_{\pm.042}$ | $.255_{\pm.005}$ | $5.837_{\pm.036}$ | $.250_{\pm.006}$ | $1.573_{\pm.016}$ | $.236_{\pm.010}$ |
| | LSM | $\mathbf{6.409}_{\pm.033}$ | $\mathbf{.728}_{\pm.003}$ | $4.390_{\pm.027}$ | $.667_{\pm.004}$ | $1.595_{\pm.017}$ | $.304_{\pm.008}$ |
| | OURS | $6.491_{\pm.035}$ | $.722_{\pm.004}$ | $\mathbf{4.383}_{\pm.025}$ | $\mathbf{.673}_{\pm.004}$ | $1.549_{\pm.017}$ | $\mathbf{.321}_{\pm.008}$ |

Metrics: Mean Absolute Error, Pearson Correlation | Methods: Methods: Fully Supervised with Ours Fine-Tuned (Top), Frozen SSL Methods and Linear Probe (Bottom).

showing minimal or no overlap on most metrics. For our other baselines, WBM uses subject-aware contrastive learning and thus performs reasonably well on subject-level tasks, but struggles on the within-subject activity task. RelCon performs poorly, showing that wearable SSL methods designed for high-frequency signals may not readily transfer to our setting. LIMU-BERT performs well, trailing only `AIM_FM` and LSM. While its reconstructive objective allows for generative evaluation, its rigid masking strategy (masking all signals at a time point) makes it incapable of performing random and signal imputation, and it further fails to generalize to structured generative tasks. mTAN does reasonably well in hypertension and activity, but does worse in all other tasks, perhaps due to the difference in settings, as mTAN was originally proposed for the irregularly-sampled time-series.

The "fully-supervised" ResNet and fine-tuned (FT) `AIM_FM` are able to significantly surpass the SSL baselines that were evaluated with a "frozen embedding + linear probe" in Activity (20), but not in the other tasks. We hypothesize that this is because a simple linear probe of the embeddings proves insufficient to fully learn the nuances of 20 classes. For other tasks, the frozen learned embeddings are sufficient, resulting in comparable or slightly degraded (due to overfitting) FT results.

**Strong Scaling Performance on 40 Million Person-Hours.** Fig. 4 show that `AIM_FM` scales more effectively than LSM across 4 different dimensions: subjects, data, compute, and model capacity. `AIM_FM`'s trend indicates a more aggressive downwards slope that has yet to saturate. This directly improves upon the "saturation effect at the upper end of the compute spectrum" noted by the LSM

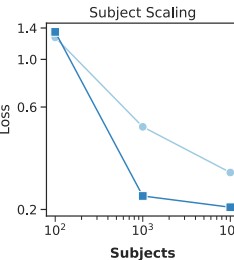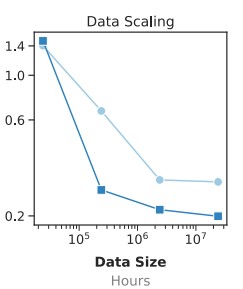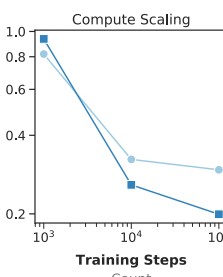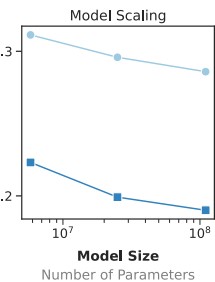

Figure 4: **Scaling Performance.** Our model achieves better scaling than LSM across all dimensions: *subjects*, *data*, *compute*, and *model size*. Our model uses a mixed masking strategy during pre-training, but here we report only random imputation loss to match LSM.

authors (Narayanswamy et al., 2024a), suggesting that this saturation is an artifact (in part) of training on imputed data. These results indicate that our model has not yet reached its fundamental limits.

**The Harm of Imputation.** Missingness is ubiquitous in wearable sensor data. Literature has shown that imputation may be finicky and introduce unintended bias (Chowdhry et al., 2021; Heymans & Twisk, 2022). Unfortunately, the practice of imputing missing data is still standard amongst SOTA wearable foundation models (Narayanswamy et al., 2024a; Erturk et al., 2025). We demonstrate that the standard practice of imputation is not only unnecessary, but is suboptimal. In Table 6, removing inheritance and forcing our model to be trained and evaluated on im-

Table 6: **Ablation study.**

| | Generative (MSE) | | Classification ($F_1$) | |
|---|---|---|---|---|
| | ↓80% Rand. Impute | ↓60m Temp. Interp. | ↑Anxiety | ↑Activity |
| AIM_FM | 0.20 | **0.45** | **0.683** | **0.474** |
| w/o Inherit | 0.28 | 0.62 | 0.671 | 0.445 |
| w/o Mixing | **0.19** | 0.58 | 0.637 | 0.460 |
| w/ Modality Specific Patch | 0.21 | 0.45 | 0.651 | 0.490 |

puted data leads to performance degradation across all of the various tasks. Furthermore, this cannot be solved by simply using "better" imputation methods. Literature has shown that stronger deep learning-based imputation methods do worse when missingness is not random (Sun et al., 2023), and additional experiments in Appendix A.4.4 show that using a more complex imputation method during pre-processing actually degrades the performance of our imputation-dependent baselines.

**Mask Mixing is an Almost Free Lunch.** AIM enables the mixing of artificial masking strategies by selecting between 80% random imputation, 50% temporal slices, or 50% signals slices for each sample. In Table 6, when mixing is ablated, a fixed 80% random masking strategy is used, matching prior work (Narayanswamy et al., 2024a). We find that ablating mask-strategy mixing degrages performance for all tasks other than random imputation, where performance is marginally affected.

**Modality Specific Encoding.** Due to the minutely aggregation, morphology-specific information that motivates modality-specific encoders may be less salient. In fact, prior work in this setting (long-context aggregated sensor features) all opted to use a single unified encoder. LSM (Narayanswamy et al., 2024a) and WBM (Erturk et al., 2025), similar to our approach, used a shared linear projection. In this way, the modality-agnostic encoder is a strength. The subsequent transformer backbone learns within- and between-modality information all together, in a parameter efficient way.

In the final row of Table 6, we modify our AIM_FM model to have modality-specific convolutional patch encoders, and the results comfirm that it does does not add significant value over the lightweight shared projection. Given these mixed results, we choose to use a shared modality-agnostic encoder, aligning our approach with the prior work and reducing the number of learnable parameters.

**Robustness to Targeted Missingness.** To simulate real-world failure modes, we evaluate AIM_FM's robustness under targeted missingness, which can be seen in Figure 5. This experiment involves two scenarios: complete sensor removal, where all features from a specific sensor (e.g., PPG) are dropped, and temporal window removal, where all sensor data from a contiguous block of time (e.g., nighttime) is removed. In these tests, AIM_FM demonstrates substantially greater resilience than the baseline LSM model. On average, AIM_FM experiences 73% smaller performance drops and retains 15% higher absolute performance across all 12 ablation settings. We investigage the utility of mask mixing and inheritance in improving robustness in Appendix A.4.3.

Crucially, AIM_FM's robust behavior is medically coherent, underscoring its reliability. For instance, hypertension and anxiety predictions show the expected nocturnal advantage, such that the removal of nighttime signals results in larger degradation than the removal of daytime. This aligns with clinical

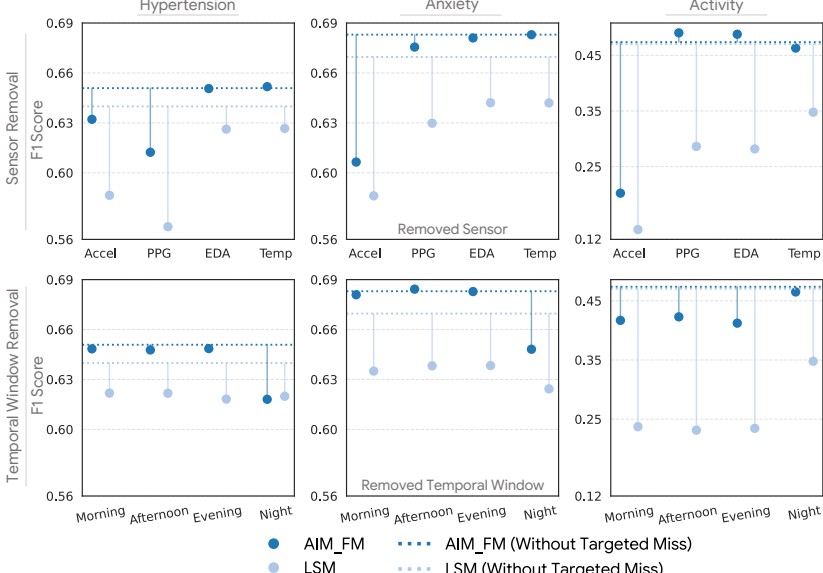

Figure 5: **Robustness to Targeted Missingness.** In sensor removal, all signals derived from the specific sensor are removed. In temporal window removal, all signals are removed at a given timeframe (Morning [8am-12pm], Afternoon [12pm-4pm], Evening [4pm-8pm], Night [8pm-8am]). The dotted line denotes optimal performance with the model trained on all data. When evaluating with simulated missingness, our method maintains consistent performance while LSM degrades significantly. Where our method does show sensitivity, it aligns with domain knowledge. For example, nighttime BP's stronger predictive power of hypertension over daytime (Hansen et al., 2011), accelerometry's role in distinguishing anxiety from physiological stress responses (Sevil et al., 2020).

literature demonstrating the diagnostic value of nighttime biosignals for hypertension (Yilmaz et al., 2023; Hansen et al., 2011) and stress prediction (Kinnunen et al., 2020; Fan et al., 2024). Additionally, our model also demonstrates a larger drop in performance for anxiety prediction after removing the accelerometry sensor compared to the other sensors. This aligns with research (Sevil et al., 2020; Wu et al., 2015) that has found that accelerometry was particularly useful for stress prediction.

**Limitations and Future Work.** *This research presents preliminary findings and should not be interpreted as providing diagnostic tools or recommendations.* Our work makes use of minutely aggregated features, useful in modeling our long context windows, but uncommon in the broader wearable sensing space, which focuses primarily on raw high frequency sensor signal. This is a practical limitation, as data is not stored in its raw form at such scale. Another limitation is the lack of validation on public data. However, as discussed earlier in the Related Work, this is because there are no public datasets that support our setting. It should be noted that although our work focuses on multimodal sensor data, `AIM` is broadly applicable and domain-agnostic. Even without inherent missingness, `AIM` can be used to efficiently handle variable and strategy-specific artificial masking ratios, a capability that standard MAEs lack. This allows us to tailor the difficulty of the pre-text task for each masking strategies respectively. For instance, we apply a high ratio for simple point-wise masking, as these tokens are easily reconstructed from local neighbors, and a lower ratio for our more structured masking, which force the model to rely on more global context. Future work should explore the application of `AIM` across different domains with variable masking ratios.

## 7 CONCLUSION

In this work, we introduced Adaptive and Inherited Masking, `AIM`, a novel self-supervised learning approach designed to learn robust representations directly from incomplete wearable sensor data. By jointly modeling inherited (real-world) and a mix of artificial masks, `AIM` eliminates the need for explicit imputation and effectively internalizes missingness in the learned representation. Using `AIM`, we train `AIM_FM`, a wearable sensor foundation model pre-trained on 40 million hours of fragmented wearable sensor data. Our experiments demonstrate that `AIM_FM` exhibits improved scaling characteristics, downstream performance, and robustness to challenging missingness scenarios. In so doing we show that missing data imputation, a standard practice for wearable FMs, is not only uneccessary but is suboptimal, a finding we hope will help inform time-series models to come.

## 8 REPRODUCIBILITY STATEMENT

We will release (1) the full metabolic study dataset (used for anxiety, hypertension, HOMA-IR, age, sex, and BMI tasks), (2) the model weights trained on this data, and (3) a codebase with the full training methodology, architecture, and reproducible evaluation code, upon acceptance.

This data was collected under informed consent in our IRB-approved study, and participants consented to data sharing under the following conditions: *"Identifiers will be removed from your identifiable private information or identifiable test results collected during this study and could then be used on its own or in combination with other data for future research studies, product development, or other commercial purposes. This data may be distributed to the Sponsor, another Investigator, affiliates, third parties, or research partners for future research studies without additional informed consent."* The ability to download our data, model weights, and software will be provided for free to qualified researchers at accredited institutions upon completion of a data use agreement.

We believe this release will provide an extremely valuable resource to the community. It will include 5.8M person-hours of wearable sensor data (241,532 day-long instances), including all derived features from the 5 wearable sensing modalities (i.e. PPG, Accelerometer, EDA, Altimeter, Temperature). While our consent language does not permit us to release the pre-training dataset of 40M hours, this approved release will expand the data available for foundation model training by several orders of magnitude. We hope that the release of this data, our AIM framework, large pre-trained models, and a unified benchmarking task will greatly accelerate the development of reproducible wearable foundation models from real-world sensor data.

## 9 ETHICS STATEMENT

While consumer health research holds potential for significant positive impact, with so many possible stake holders, such research must be performed intentionally to ensure that it is safe and fair. Additionally, there exists the unfortunate possibility that bad-actors may attempt to leverage methods, such as our own, in negligent ways. As researchers in the field, the burden falls to us to consider the implications of this research, and act to fulfill the positive impacts and mitigate the associated risks. Additionally, we note that we have used LLMs to help edit and polish writing within this submission to help rewrite specific phrases and assist in framing ideas in a way that reflected the authors' original intent.

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

## Appendix Table of Contents

## A.1 DATA DETAILS

### A.1.1 IMPUTING MISSINGNESS FOR NON AIM MODELS

Although AIM is able to organically handle existing missing values using clever masking, the same cannot be said for our baseline methods. Furthermore, many standard deep learning frameworks (such as pytorch, jax, and tensorflow) are unable to handle nan values in model training and evaluation, causing value errors or propogating nans throughout the network during forward and backward passes. For this reason we impute missing (nan) values in our data. We use linear interpolation between gaps and then back and forward fill for missingness at the start and end of the sequence.

### A.1.2 DEVICE DETAILS

There are many different types of smartwatches and fitness trackers. Fig. 6 shows the distribution of different trackers and smartwatches present in our pretraining dataset. Given the scale of our dataset we are able to train on examples of data from many different devices. Consequently, our model demonstrates robustness across diverse device types, handling their varying sensor technologies and differing inherent missingness patterns.

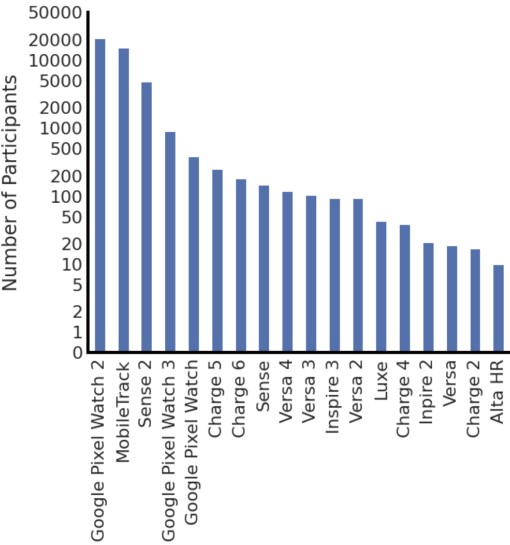

Figure 6: **Device Distribution.** The count of each fitness tracker present in our pre-training dataset.

### A.1.3 SENSOR DERIVED MINUTELY FEATURES

Our wearable devices utilize 5 different sensors: Photoplethysmography, Accelerometer, Skin Conductance (electrodermal activity or EDA), Temperature, and Altitude. Each of these sensors collects raw waveform signals at 100 Hz, 25 Hz, 200 Hz, 6 Hz, amd 10 Hz respectively, but we do not use the signals at this high resolution because (1) due to practical reasons (i.e. prohibitive storage costs and battery drain), data is not stored in this raw form at our scale, and (2) it is computationally impractical to learn models on raw waveforms across an entire day (i.e. 200 Hz for 1 day is $T = 17$ million time-points, per instance). As such, various features are curated from the raw waveforms as minutely aggregated features and saved to be used as inputs into our model. Each of these features are grounded in the domain literature, based on prior work that has shown their clinical effectiveness. For example, heart rate variability metrics like RMSSD (DeGiorgio et al., 2010) or Shannon Entropy of RR intervals (Afdala et al., 2017) have well-established prognostic value for cardiovascular health, while accelerometry features like jerk ratio (Pan et al., 2020) effectively characterize movement quality.

Each of the derived features, as well as their base sensor origin, can be found in Table 7 below. For the targeted sensor removal experiments, as well as any other descriptions of the sensor as a whole,

*we refer to the sensor as all features derived from the sensor.* For example, when removing the PPG sensor in the targetted missingness experiment, we remove all PPG-derived features, from Heart Rate to Shannon Entropy RR Differences.

Table 7: **Sensor Feature Definitions and the Sensor they are Derived From.**

| Feature | Unit | Definition |
|---|---|---|
| **Photoplethysmography** | | |
| Heart Rate | Beats/Min | Mean of instantaneous heart rate. |
| Heart Rate at Rest | Beats/Min | Mean of heart rate at rest. |
| RR Percent Valid | % | % of 5-minute window with valid RR intervals. |
| RR $80^{th}$ Percentile | Msec | $80^{th}$ percentile of 5-minute window of RR ints. |
| RR $20^{th}$ Percentile | Msec | $20^{th}$ percentile of RR ints. |
| RR Median | Msec | Median RR interval. |
| RMSSD | Msec | Root mean squared st. dev. of RR ints. |
| SDNN | Msec | Standard deviation of RR intervals. |
| Shannon Ent. RR | Nats | Shannon entropy of the RR intervals. |
| Shannon Ent. RR Diffs | Nats | Shannon entropy of the RR interval differences. |
| **Accelerometer** | | |
| Step Count | Steps | Number of steps. |
| Jerk Autocorrelation Ratio | a.u. | Ratio of lag=1 autocorrelation to energy in 1st 3-axis principal component. |
| Log Energy | a.u. | Log of sum of 3-axis root mean squared magnitude. |
| Covariance Condition | a.u. | Estimate of condition number for the 3-axis covariance. |
| Log Energy Ratio | a.u. | Log of ratio of sum of energy in 1st 3-axis principal component over energy of 3-axis root mean squared magnitude. |
| Zero Crossing St.Dev. | Seconds | Standard deviation of time between zero crossing of 1st 3-axis principal component. |
| Zero Crossing Average | Seconds | Mean of time between zero crossing of 1st 3-axis principal component. |
| Axis Mean | a.u. | Mean of 3-axis |
| Kurtosis | a.u. | Kurtosis of 3-axis root mean squared magnitude. |
| Sleep Coefficient | a.u. | Sum of 3-axis max-min range with 16 log-scaled bins. |
| **Skin Conductance** | | |
| Skin Conductance Value | $\mu$Siemens | Center of linear tonic SCL value fit. |
| Skin Conductance Slope | $\mu$S/Min | Intraminute slope of SCL values. |
| Lead Contact Counts | Counts | Number of times sensor leads contacted the wrist in a minute. |
| **Skin Temperature** | | |
| Skin Temperature Value | $^{\circ}$C | Mean value of skin temperature. |
| Skin Temperature Slope | $^{\circ}$C/Min | Slope of skin temperature. |
| **Altimeter** | | |
| Altitude St.Dev. Norm | Hectopascals | Standard deviation of altimeter readings. |

### A.1.4    DEMOGRAPHIC BREAKDOWN

A statistical breakdown of our datasets, by demographic features can be found in Table 8. A subset of these, age and BMI, represent two of the regression tasks used to validate our method.

### A.1.5    DISCRIMINATIVE TASK LABEL BREAKDOWN

Table 9 shows label and data breakdown of the discriminative tasks used to validate our method. These tasks include 20-class activity recognition (Table 9(a)) from the activity dataset, and binary anxiety and hypertension classification (Table 9(b.i)) from the metabolic dataset.

### A.1.6    ACQUISITION AND APPROVAL

The data used for training in our analysis was curated from a large corpus of historical wearable data collected with consent from partcipants for these data to be used in research. Specifically, the

Table 8: **Demographics of our Various Datasets.**

| Category | Pre-training | | Downstream Activity | | Downstream Metabolic | |
|---|---|---|---|---|---|---|
| | Train (%) | Val (%) | Train (%) | Val (%) | Train (%) | Val (%) |
| **Sex** | | | | | | |
| Male | 37,352 (68.1) | 3,657 (63.8) | 27,653 (73.1) | 6,092 (73.0) | 551 (44.1) | 258 (35.4) |
| Female | 23,041 (38.1) | 2,065 (36.0) | 10,145 (26.8) | 2,248 (26.9) | 670 (53.6) | 455 (62.4) |
| Not Specified | 48 (0.1) | 10 (0.2) | 24 (0.1) | 3 (0.1) | 0 (0) | 0 (0) |
| **Age** | | | | | | |
| 18–39 | 28,519 (47.2) | 2,583 (45.1) | 19,340 (51.1) | 4,492 (53.8) | 415 (33.2) | 223 (30.6) |
| 40–59 | 24,888 (41.2) | 2,433 (42.4) | 15,309 (40.5) | 3,172 (38.0) | 637 (51.0) | 384 (52.7) |
| 60–79 | 6,473 (10.7) | 664 (11.6) | 2,875 (7.6) | 618 (7.4) | 198 (15.8) | 121 (16.6) |
| $\geq$80 | 364 (0.6) | 39 (0.7) | 120 (0.3) | 31 (0.4) | 0 (0) | 1 (0.1) |
| Not Specified | 197 (0.3) | 178 (0.5) | 30 (0.4) | 0 (0) | 0 (0) | 0 (0) |
| **BMI** | | | | | | |
| Healthy ($<$25) | 22,425 (37.1) | 2,173 (37.9) | 15,942 (42.2) | 3,685 (44.2) | 319 (25.5) | 188 (25.8) |
| Overweight (25–30) | 20,242 (33.5) | 1,952 (34.1) | 14,154 (37.4) | 3,017 (36.2) | 343 (27.4) | 206 (28.6) |
| Obese ($\geq$30) | 14,799 (24.5) | 1,330 (23.2) | 6,131 (16.2) | 1,316 (15.8) | 481 (38.5) | 274 (37.6) |
| Not Specified | 230 (0.4) | 14 (0.2) | 81 (0.2) | 18 (0.2) | 49 (3.9) | 28 (3.8) |
| **Total** | 60,440 (100) | 5,732 (100) | 37,822 (100) | 8,343 (100) | 1,250 (100) | 729 (100) |

Table 9: **Discriminative Task Dataset Distribution**

(a) **Activity Recognition Dataset**

| Task / Label | Train (%) | Test (%) |
|---|---|---|
| **Activity** | | |
| Walk | 4,434 (6.0) | 874 (5.8) |
| Bike | 4,363 (5.9) | 858 (5.6) |
| Sport | 4,433 (6.0) | 902 (5.9) |
| Run | 4,023 (5.4) | 790 (5.2) |
| Aerobics | 4,417 (6.0) | 906 (6.0) |
| Elliptical | 4,402 (5.9) | 879 (5.8) |
| Spinning | 4,402 (5.9) | 858 (5.6) |
| Weightlifting | 4,335 (5.9) | 841 (5.5) |
| Swim | 4,280 (5.7) | 867 (5.8) |
| Hike | 4,062 (5.5) | 841 (5.5) |
| Tennis | 4,138 (5.6) | 815 (5.4) |
| CrossFit | 4,305 (5.8) | 887 (5.8) |
| Pilates | 4,365 (5.9) | 846 (5.6) |
| Stairclimber | 4,272 (5.8) | 834 (5.5) |
| Dancing | 4,288 (5.8) | 826 (5.4) |
| Indoor climbing | 3,520 (4.8) | 853 (5.6) |
| Golf | 3,003 (4.1) | 710 (4.7) |
| Skiing | 1,594 (2.1) | 420 (2.8) |
| Snowboarding | 662 (0.9) | 167 (1.1) |
| Kayaking | 732 (1.0) | 212 (1.4) |
| **Total** | 74,030 (100) | 15,186 (100) |

(b.i) **Metabolic Dataset** Classification Tasks

| Task / Label | Train (%) | Test (%) |
|---|---|---|
| **Anxiety** | | |
| Positive | 55,030 (36.4) | 34,749 (38.5) |
| Negative | 96,316 (63.6) | 55,437 (61.5) |
| **Hypertension** | | |
| Positive | 36,349 (24.0) | 23,353 (25.9) |
| Negative | 114,997 (76.0) | 66,833 (74.1) |
| **Total** | 151,346 (100) | 90,186 (100) |

consent language described use of the data for developing new health features and algorithms and being included in publications:

*REDACTED will collect and use your data to research and develop new health and wellness products and services for you and others. This data includes your: Health and wellness data, such as steps, heart rate, and sleep data. Your data may also be used to generate findings that could be included in publications (such as scientific journals) to contribute to general knowledge about health and science. For example, activity, heart rate, and sleep data contributed to published findings that Fitbit devices could help detect flu outbreaks. None of the data used for these purposes will include your name, email, or other information that directly identifies you.*

The use of data for pretraining in this manner was approved as exempt under 45 CFR § 46.104(d)(4) *"because the research involves the use of identifiable private information/biospecimens; and information, which may include information about biospecimens, is recorded by the investigator in such a manner that the identity of the human subjects cannot readily be ascertained directly or through identifiers linked to the subjects, the investigator does not contact the subjects, and the investigator will not re-identify subjects."*

The Metabolic downstream dataset for anxiety and hypertension prediction came from an IRB approved study (protocol number removed for anonymization). The core objective of this study as described in the IRB protocol was to: *"Evaluate the feasibility of using the data provided by wrist-worn wearable devices to develop algorithms and scores to assess metabolic health."*

In the consent for the observational study, participants were informed that data on up to 7,500 participants in the United States would be collected. We used a mobile study platform that allows participants to enroll, check eligibility and provide full informed consent. The same mobile application enables the collection of Fitbit data using Fitbit devices or Pixel watches and allows participants to complete questionnaires. The participants reported their anxiety, depression and hypertension diagnoses through this app. Data was de-identified and stored in accordance with the approved IRB protocol. The participants were compensated with a free set of lab tests from Quest Diagnostics for participating in the study.

## A.2 MISSINGNESS VISUALIZATIONS

A core property of these data is that they are fragmented, and the missingness has several modal types. Three very common modes occur: 1) When the device is being charged or off all sensor stop recording data (device off), 2) when the device is in certain operation modes (e.g., when in sleep mode) certain signals stop being recorded (sensor off) and 3) when there is noise in the sensor data spurious values (e.g., values that are not physiologically possible - HR=0) are filtered out. The following sections demonstrate additional visualizations of the missingness patterns present from these mechanisms.

### A.2.1 ADDITIONAL EXAMPLES OF DATA WITH EXISTING MISSINGNESS

In order to demonstrate the ubiquity and broad range of missingness patterns found within the data, we randomly sample an additional 8 data examples, shown in Fig. 7. These examples further demonstrate how some patterns are consistent across users, such as increased missingness during early morning hours (12am-6am) (reflecting device removal during sleep) or correlated missingness dropout across various sensor channels. However, it should be noted that all samples exhibit unique missingness signatures with no two patterns being identical with vastly differing missingness percentages (27-63%) and demonstrating the ubiquity of real-world missingness. These findings motivated our development of AIM's flexible masking approach, which explicitly models such heterogeneous missingness patterns during pre-training.

### A.2.2 PREVALENCE AND LENGTH OF MISSINGNESS

In Fig. 8, we demonstrate the prevalence of missingness as well as the length of the missingness, broken down across each sensor type across all 1.6 million instances of pre-training data. As we can see, each sensor has very different patterns of missingness, and across all sensors, their missingness presents as long extended gaps, making them non-trivial to reconstruct over. Notably, the

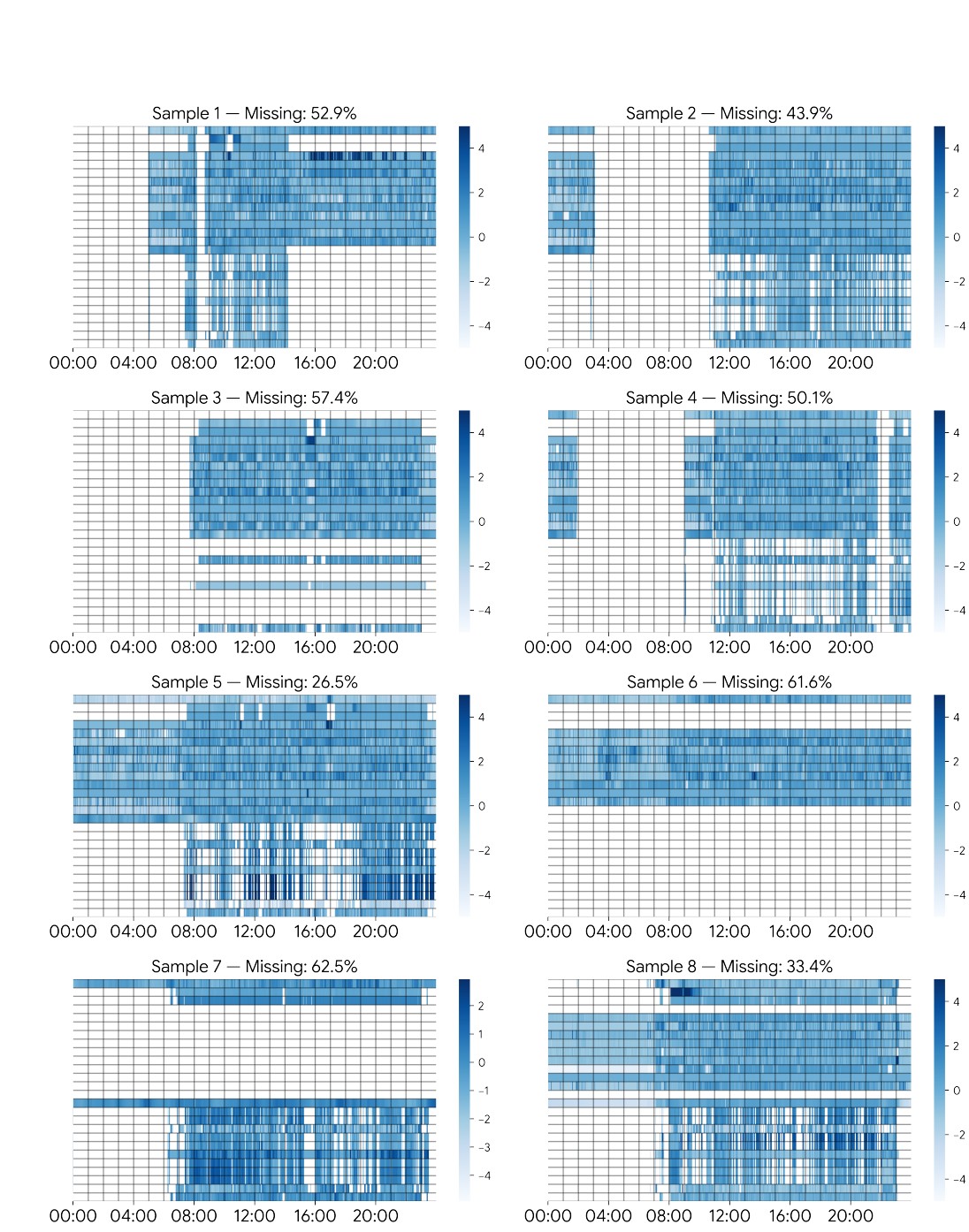

Figure 7: **Gallery of Data Examples with Real-world Missingness.** White designates missingness.

accelerometry features in particular, have missingness in the form of these extended gaps, whereas most of the missingness for PPG sensors is of shorter length.

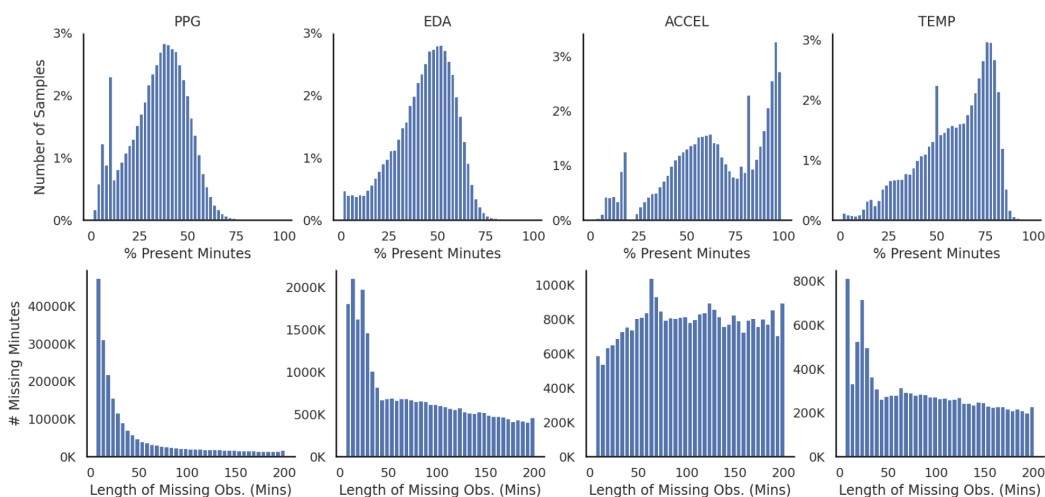

Figure 8: **Distribution of Prevalence and Length of Missingness.**

## A.3 Model Hyperparameter and Implementation Details

### A.3.1 Pre-training Set-up.

We pre-train our models on a large set of wearable minutely sensor data described. The raw multimodal sensor data input can be denoted by $\mathbf{A} \in \mathbb{R}^{T \times S}$. $S = 26$, which is the full number of signals in our multimodal data. These signals are derived from 4 different wearable sensors: Accelerometry, PPG, EDA, and Temperature. In our setting, we set $T = 1440$, which is composed of all minutes from a full 24 hour day, from midnight to midnight local time. We use this window size as days normally have a consistent structure, allowing for a more meaningful absolute positional embedding than if an arbitrary window size was set (e.g. 300 minutes (Narayanswamy et al., 2024a)).

Our model was pre-trained with a ViT-1D (Dosovitskiy et al., 2020; Abbaspourazad et al., 2023) encoder backbone by using a 1D patch size of 10 time-steps (i.e. 10 minutes). This results in a total of 3744 tokens (the 1440 minutes are reduced to 144 tokens per signal. With 26 signals, 26*144=3744 is the final number of tokens). Similar to prior work (Na et al., 2024), each signal channel is patched with a shared kernel, and we utilize a 2D positional embedding to encode information about the temporal position and signal channel. The ViT model had 25 million parameters with an encoding dimensionality of 384, 12 encoder layers, and 4 decoder layers. Our mask is a union of the inherited mask with an artificial masking mix of 80% random imputation, 50% temporal slices, and 50% signal slices. Our primary pre-training objective is to optimize the signal reconstruction loss (i.e. mean squared error), averaged over the artificially masked patches. The model was pre-trained on 8x16 Google v5e TPUs with a total batch size of 512 across 100,000 training steps. The training process uses the AdamW optimizer with a base learning rate of $5e-3$, weight decay set to $1e-4$, and betas set to 0.9 and 0.95. Gradients were clipped at 1.0. A linear warm-up schedule is applied for the first 5% of total steps, followed by a cosine learning rate decay to zero.

Our SSL baselines include LSM (Narayanswamy et al., 2024a), SimCLR (Chen et al., 2020), DINO (Caron et al., 2021), and a Masked Siamese Network (MSN) (Assran et al., 2022). LSM is an MAE (He et al., 2022) approach with 0.8 random masking ratio with no inherited masking. SimCLR, DINO, and MSN are augmentation-based contrastive approaches, and we utilize a set of common time-series augmentations (Tang et al., 2020; Liu et al., 2024; Zhang et al., 2022; Rommel et al., 2022): jittering, scaling, and time flipping. Each augmentation has a 0.5 probability of being applied. Jittering was implemented as a random sample from a gaussian distribution with zero-mean and a uniformly randomly sampled standard deviation frp, 0 to 0.5, per value in the time-series. Scaling was implemented by multiplying all of the data input with a scale, uniformly sampled from 1.1 to 1.5. For DINO, we omit scaling as the model was unable to converge.

Each of these baselines were all pre-trained from scratch, following the same previously stated training conditions, unless stated otherwise. All baselines expect full, complete data as input, and as such, they utilize the imputed version of our sensor dataset. LSM was trained with a ViT-2D with a 2D patch size of (10,2), in order to match their image-based encoding approach, and all other ViT parameters remain constant.

### A.3.2 Downstream Evaluation

We group our downstream evaluation into three sections based on the target: generative, classification, and regression.

In our **Generative Evaluation**, we evaluate how well our model is able to reconstruct different types of structured missingness patterns that mimic real-world missingness patterns: (1) Random Imputation, where a [30%, 50%, 80%] of tokens is masked out, (2) Temporal Interpolation, where all signals in a contiguous temporal window of length [10, 30, 60 minutes] is completely masked out, (3) Temporal Extrapolation, which is similar to interpolation, but the window is necessary at the end of the time-series, and (4) Signal Imputation, where all time points for a random set of [2/26, 6/26, 12/26] signal channels is masked. Reconstruction performance was calculated with mean squared error (MSE) on the artificially masked tokens, averaging only over the data points that have a ground truth.

Our deep learning baselines include the LSM model (Narayanswamy et al., 2024a), another MAE-based model, which can be used to evaluate these generative tasks out-of-box by setting the artificial

masking procedure to match the proposed tasks. Our `AIM` model is done in the same way, but the full encoder mask includes the inherited mask as well. Unfortunately, the contrastive SSL baselines are unable to provide generative performance metrics because they do not utilize a reconstruction objective. Instead, we use alternative simple generative baselines, which match practical applications. Many application-focused biosensor algorithms will employ simple imputation methods (Pires et al., 2020; Xu et al., 2022; Srimedha et al., 2022; Wu et al., 2020; Amiri & Jensen, 2016) as quick data preprocessing methods. Thus, we choose to include these additional methods as baselines: Linear Interpolation, K-Nearest Neigbhors, and Mean Filling. Similar to our method, we run these baselines with a union mask of the mask inherited from existing missingness and the artificial mask. MICE (Van Buuren & Groothuis-Oudshoorn, 2011) is another popular, simple baseline designed for multivariate data, but we opted to not include it due to our existing missingness patterns violating the Missingness At Random assumption, and prior work demonstrate a relative poorer performance compared to nearest neighbor and linear interpolation (Narayanswamy et al., 2024a).

In our **Classification Evaluation**, we evaluate how well our model's embedding representation is able to capture discriminative features. During evaluation, our model calculates the embedding on all non-inherited-masked tokens and uses an average pooling followed by a trainable linear probe to classify each of the prediction targets. For the LSM model, because it is unable to represent the inherited mask, the embedding for all tokens is pooled, such that tokens that were part of the existing missingness but have been filled with imputation will be included. For the contrastive methods, the learned CLS token is used as the pooled representation. We report performance with F1 score as it balances precision and recall for class-imbalanced targets, Accuracy as a straightforward measure of overall correctness, Balanced Accuracy to account for potential class imbalance, and AUROC to evaluate the model's ranking capability across all classification thresholds. The prediction targets are hypertension, anxiety, which originate from the Metabolics dataset and 20-class activity recognition, which originates from the Activity dataset.

The linear probe was trained by freezing the learned ViT backbone, averaging over the entire embedding and training a logistic regression head ontop of it. For our `AIM` model specifically, with the inherited mask, the average was only done over the non-masked tokens. Training was done with a batch size of 512, across 500 training steps with an AdamW optimizer with a base learning rate of $5e-3$, weight decay set to $1e-4$, and betas set to 0.9 and 0.95. Gradients were clipped at 1.0. For activity specifically, training steps and learning rate were increased to 1000 and $1e-1$ to achieve better convergence.

Additionally, we include two extra supervised baselines, ViT-1D (Dosovitskiy et al., 2020) and a ResNet (He et al., 2016), that are trained end-to-end for each of our tasks. ViT-1D is a transformer-based architecture that follows the same architecture as our `AIM` with 25 million parameters, but with randomly initialized weights, trained end-to-end. ResNet is a strong CNN-based architecture that has seen broad success throughout the health biosignal time-series domain (Xu et al., 2024; Pillai et al., 2025; Abbaspourazad et al., 2023; Mekruksavanich et al., 2022). This model was a ResNet-50 (He et al., 2016) with 25 million parameters, in order to match the ViT model. Specifically, it contains 50 layers, with 64 filters that double after each residual block, with a final average pooling and logistic regression head. Both models are trained with a batch size of 512, across 500 training steps with an AdamW optimizer with a base learning rate of $5e-3$, weight decay set to $1e-4$, and betas set to 0.9 and 0.95. Gradients were clipped at 1.0. A linear warm-up schedule is applied for the first 5% of total steps, followed by a cosine learning rate decay to zero. Because these models do not handle missingness, they were trained directly on the imputed data.

In our **Regression Evaluation**, we utilize the same evaluation procedure described in classification, only instead the linear probe is specifically a linear regression. We report performance with MAE as it provides an interpretable deviation from the correct value, as well as Pearson Correlation Coeffecient, as it is a common metric for evaluating how well a regressor is able to capture the trend of the target (Xu et al., 2024; Yuan et al., 2024). The prediction targets are BMI and Age.

The linear probe was trained by freezing the learned ViT backbone, averaging over the entire embedding and fit a linear regression head ontop of it using Scikit-Learn's LinearRegression implementation out-of-box. The supervised baselines were trained in an identical way as done in the classification evaluation, but using a linear regression head instead of logistic regression.

## A.4  ADDITIONAL RESULTS

### A.4.1  CONFUSION MATRICES

Fig. 9 illustrates the utility of `AIM` learned embeddings for downstream applications. Specifically, this confusion matrix shows the performance of `AIM`, post-trained on the 20-class activity recognition task using a linear probe. It is clear that the embedding are useful in discriminating between a large number of activities, even those which may be semantically clustered, such as skiing and snowboarding. Future work may explore how to expand to even more activities and behavioral events, and investigate the utility of large-scale pre-training in address long-tail task labels.

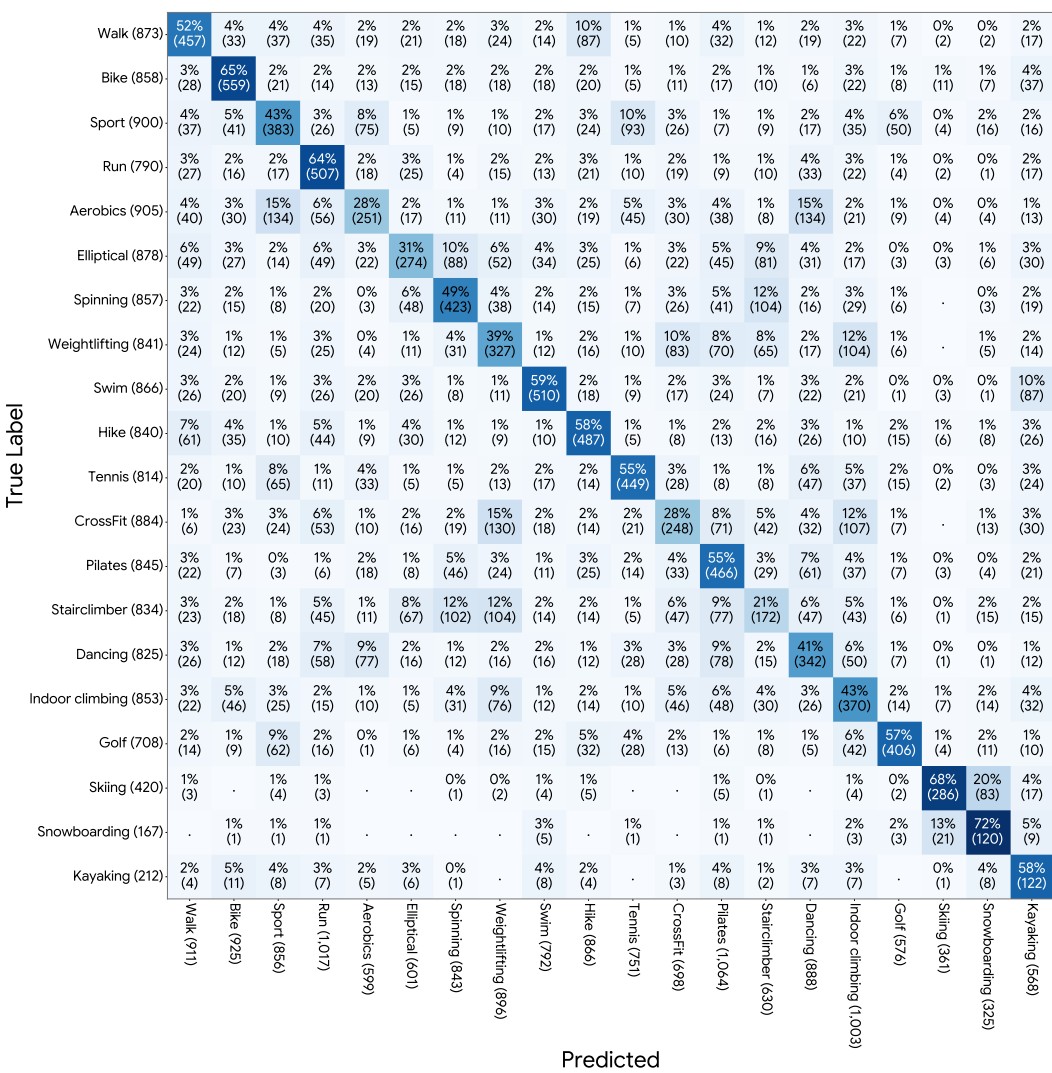

Figure 9: **Activity Recognition Confusion Matrix.** The results of a linear probe applied to `AIM` for the 20-class activity recognition task. Rows add up to 100%.

### A.4.2  RECONSTRUCTION EXAMPLES

Fig. 10 shows various reconstruction examples for a specific sensor signal. Here we can clearly see Our `AIM` approach leads to much stronger performance, across different generative tasks.

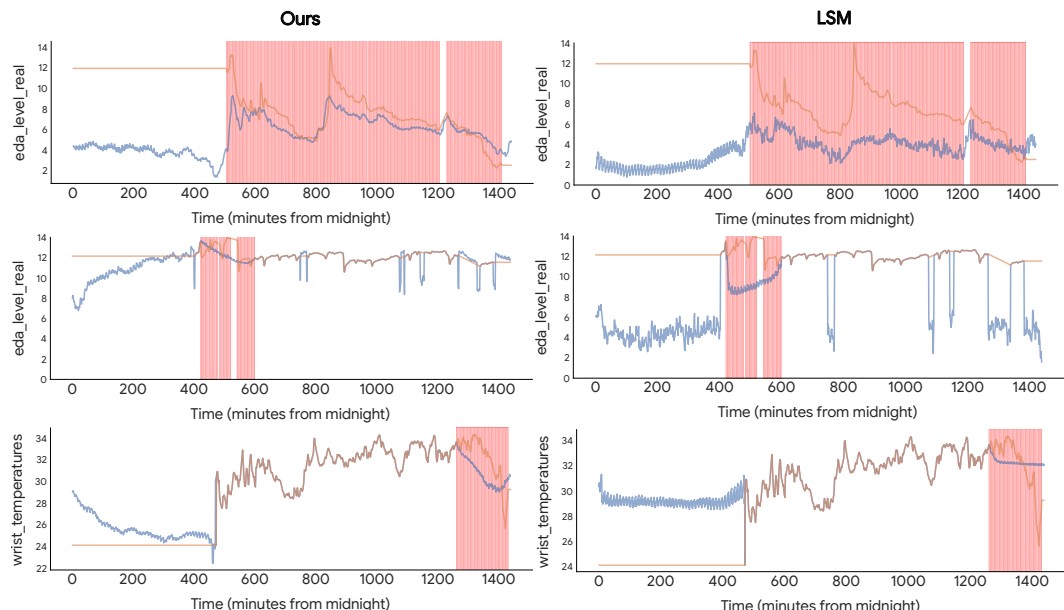

Figure 10: **Reconstruction Examples for 2/26 Sensor Signal Imputation (Row 1), 3 Hour Temporal Interpolation (Row 2), 3 Hour Temporal Extrapolation (Row 3).** Red highlighted regions demonstrate regions of artificial masking. Orange shows original data with imputation (i.e. the first 400-500 steps of the each row were originally missing, then imputed, as demonstrated by the straight line) and blue shows the reconstructed data.

### A.4.3 DISENTANGLING ROBUSTNESS

To disentangle the benefits of inheritance versus our masking mix, we conducted a new ablation study. We fully pre-trained and evaluated two ablated models:

- InheritOnly: Uses inherited masking, but with only a simple 80% random artificial mask.
- MixOnly: Uses our diverse mix of artificial masks (80% random, 50% temporal, and 50% signal-slice), but no inherited masking.

The results of this robustness evaluation, with 95% CIs from 100 bootstrap iterations, are shown in Table 10. "No rm" denotes baseline performance, while subsequent rows show performance after targeted data removal.

Table 10: **Ablation with Robustness Experiment**

| Method | Hypertension | | Anxiety | | Activity | |
|---|---|---|---|---|---|---|
| | InheritOnly | MixOnly | InheritOnly | MixOnly | InheritOnly | MixOnly |
| No rm | $0.637 \pm 0.003$ | $0.644 \pm 0.003$ | $0.663 \pm 0.003$ | $0.671 \pm 0.003$ | $0.460 \pm 0.008$ | $0.445 \pm 0.007$ |
| rmACC | $0.622 \pm 0.003$ | $0.614 \pm 0.004$ | $0.598 \pm 0.004$ | $0.603 \pm 0.004$ | $0.230 \pm 0.006$ | $0.243 \pm 0.006$ |
| rmPPG | $0.601 \pm 0.003$ | $0.609 \pm 0.003$ | $0.662 \pm 0.003$ | $0.661 \pm 0.003$ | $0.477 \pm 0.009$ | $0.393 \pm 0.008$ |
| rmEDA | $0.638 \pm 0.003$ | $0.645 \pm 0.003$ | $0.661 \pm 0.003$ | $0.670 \pm 0.003$ | $0.478 \pm 0.008$ | $0.445 \pm 0.008$ |
| rmTEMP | $0.633 \pm 0.003$ | $0.644 \pm 0.003$ | $0.662 \pm 0.003$ | $0.667 \pm 0.003$ | $0.455 \pm 0.009$ | $0.427 \pm 0.008$ |
| rmNight | $0.615 \pm 0.003$ | $0.621 \pm 0.004$ | $0.641 \pm 0.003$ | $0.635 \pm 0.003$ | $0.453 \pm 0.009$ | $0.403 \pm 0.008$ |
| rmMorning | $0.635 \pm 0.003$ | $0.643 \pm 0.004$ | $0.661 \pm 0.003$ | $0.669 \pm 0.003$ | $0.404 \pm 0.008$ | $0.381 \pm 0.007$ |
| rmAfternoon | $0.634 \pm 0.003$ | $0.643 \pm 0.004$ | $0.664 \pm 0.003$ | $0.671 \pm 0.003$ | $0.416 \pm 0.008$ | $0.388 \pm 0.007$ |
| rmEvening | $0.633 \pm 0.003$ | $0.639 \pm 0.003$ | $0.662 \pm 0.004$ | $0.670 \pm 0.003$ | $0.396 \pm 0.008$ | $0.371 \pm 0.008$ |

Metrics: $F_1$ Score

The results reveal a clear and informative trade-off. InheritOnly is significantly more robust for Activity recognition, maintaining higher performance across most removal scenarios. The one

exception is a large performance drop when accelerometry data is removed (rmACC). This is an expected and desirable outcome, as it confirms the model correctly learns to depend on this key modality. Conversely, MixOnly more consistently outperforms InheritOnly on the Hypertension and Anxiety prediction tasks, achieving higher baseline scores and retaining its advantage after most removals.

We hypothesize that this phenomena occurs due to the differences in local versus global temporal information. In our day-long time-series, activity recognition needs to isolate local temporal information, and thus, inheritance is useful for helping the model identify specific local missingness structures that are more systematically related to the activity and the behavior of the device (e.g., motion artifacts). Hypertension and anxiety are global, subject-level states that require synthesizing information over a full day. For this, our masking mix is more beneficial, as its temporal and signal-slice strategies explicitly train the model to reconstruct long-range context, handling the loss of entire modalities and incorporating data that is not specific to any one activity; this is probably more useful for forming global representations.

### A.4.4  STRONGER IMPUTATION MAY INSTEAD HURT PERFORMANCE

Prior works have found that stronger imputation methods may introduce unintended bias [1, 2], a plausible reason as to why simple imputation remains the standard approach (Goyal et al., 2019; Erturk et al., 2025). To test this, we re-train + re-evaluate all baselines with quadratic interpolated data and compare it against mean imputed data. Evaluations cover 5 classifcation and regression tasks derived from our metabolic study dataset.

Table 11: **Downstream Performance after Re-training and Re-evaluating with Quadratic Interpolation**. Numbers indicate the performance after re-train and re-evaluating. The amount of degradation, compared to the results with mean imputation are shown in the (). If this was a performance loss, then it is marked with an underline.

| Method | Hypertension (2) | | | | Anxiety (2) | | | | Age (R) | | BMI (R) | | Inslin Resis. (R) | |
|---|---|---|---|---|---|---|---|---|---|---|---|---|---|---|
| | ↑$F_1$ | ↑Acc | ↑BAcc | ↑AUC | ↑$F_1$ | ↑Acc | ↑BAcc | ↑AUC | ↓MAE | ↑Corr | ↓MAE | ↑Corr | ↓MAE | ↑Corr |
| ResNet | 0.49(−0.04) | 0.58(−0.01) | 0.49(−0.03) | 0.62(−0.01) | 0.63(−0.03) | 0.64(−0.01) | 0.63(−0.02) | 0.70(−0.01) | 7.58(0.15) | 0.60(−0.02) | 5.19(0.12) | 0.49(−0.02) | 1.59(−0.05) | 0.24(0) |
| ViT-1D | 0.42(−0.10) | 0.51(0) | 0.42(−0.06) | 0.53(0.01) | 0.59(0) | 0.58(−0.01) | 0.57(−0.01) | 0.61(−0.01) | 9.75(0.09) | 0.08(−0.05) | 5.97(−0.09) | 0.04(−0.01) | 1.62(0.03) | 0.07(−0.07) |
| LIMU-BERT | 0.60(0) | 0.60(0) | 0.56(0) | 0.63(−0.01) | 0.63(−0.01) | 0.64(−0.01) | 0.63(−0.01) | 0.69(−0.01) | 8.61(0.17) | 0.44(−0.03) | 5.58(0.09) | 0.38(−0.03) | 1.61(0.01) | 0.21(−0.02) |
| BSD | 0.59(0.01) | 0.59(0.02) | 0.55(0.01) | 0.62(0.02) | 0.63(0.03) | 0.64(0.03) | 0.63(0.03) | 0.68(0.04) | 8.56(−0.05) | 0.46(0.01) | 5.61(−0.05) | 0.36(0.01) | 1.60(−0.11) | 0.25(0.02) |
| RelCon | 0.54(−0.03) | 0.54(−0.02) | 0.51(−0.02) | 0.56(−0.03) | 0.61(0) | 0.61(0) | 0.60(0) | 0.65(0) | 9.03(0.14) | 0.37(−0.01) | 5.80(0.27) | 0.26(−0.15) | 1.59(−0.02) | 0.24(0.05) |
| SimCLR | 0.55(0.02) | 0.57(0.02) | 0.52(0.02) | 0.59(0.02) | 0.60(0.08) | 0.60(0.05) | 0.60(0.09) | 0.64(0.07) | 9.12(−0.09) | 0.35(0.01) | 5.84(−0.01) | 0.22(−0.01) | 1.56(0.02) | 0.22(−0.03) |
| Dino | 0.53(−0.01) | 0.53(0.03) | 0.50(0.01) | 0.55(0.04) | 0.58(0.04) | 0.58(0.08) | 0.57(0.09) | 0.61(0.10) | 9.40(−0.29) | 0.25(0.13) | 5.90(−0.07) | 0.19(0.07) | 1.58(−0.01) | 0.18(0.09) |
| MSN | 0.52(−0.04) | 0.52(−0.03) | 0.49(−0.03) | 0.53(−0.04) | 0.57(0.02) | 0.57(0.02) | 0.56(0.04) | 0.60(0.02) | 9.58(0.16) | 0.19(−0.06) | 5.95(0.11) | 0.17(−0.08) | 1.60(0.03) | 0.14(−0.10) |
| LSM | 0.57(−0.11) | 0.57(−0.11) | 0.54(−0.11) | 0.59(−0.15) | 0.61(−0.07) | 0.61(−0.07) | 0.60(−0.04) | 0.65(−0.09) | 9.13(2.72) | 0.35(−0.38) | 5.77(1.38) | 0.28(−0.39) | 1.57(−0.02) | 0.24(−0.07) |
| Ours (No Impute) | **0.69** | **0.69** | **0.65** | **0.75** | **0.69** | **0.69** | **0.68** | **0.76** | **6.49** | **0.72** | **4.38** | **0.67** | **1.55** | **0.32** |

The results indicate that more complex imputation was largely detrimental. 6/9 baselines had performance degradation on at least 10/12 metrics across the tasks. SimCLR and WBM, showed mixed improvements with performance still degraded on the Hypertension and HOMA-IR tasks, respectively. Crucially, no baseline was able to achieve better performance than our model, which does not require imputation. We conclude that poor baseline performance cannot be primarily attributed to naive imputation strategies.

### A.4.5  THE UTILITY OF MODALITY SPECIFIC MODELING

By nature different sensor modalities encode complementary information about physiology and behavior. For example a PPG sensor likely better encodes cardiovascular health as compared to an IMU which may focus on motion features. As such, explicitly enabling the model to learn these sensor-specific nuances may affect performance. To this end we conduct an ablation study to determine if sensor-specific modeling improves overall performance. Specifically, we adapt our model to include a sensor-specific patch-encoder for each of the 5 sensor modalities (PPG, IMU, EDA, Skin Temperature, Altimeter).

We find that such sensor-specific modeling results in very similar performance to that of `AIM_FM`. Specifically, we find that modality specific encoding slightly improves performance for temporal interpolation and extrapolation, while exhibiting slight degradation for random imputation and signal imputation. Modality specific encoding slightly improves performance on the hypertension and

Table 12: **The Utility of Modality Specific Modeling.**

**Generative**

| Method | Random Imp. (MSE) | | | Temporal Interp. (MSE) | | | Temporal Extrap. (MSE) | | | Signal Imp. (MSE) | | |
|---|---|---|---|---|---|---|---|---|---|---|---|---|
| | 30% | 50% | 80% | 10m | 30m | 60m | 10m | 30m | 60m | 2 | 6 | 12 |
| OURS + Modality Specific Enc. | $0.119_{\pm.000}$ | $0.138_{\pm.001}$ | $0.223_{\pm.002}$ | $\mathbf{0.323}_{\pm.006}$ | $\mathbf{0.446}_{\pm.007}$ | $\mathbf{0.533}_{\pm.007}$ | $\mathbf{0.436}_{\pm.013}$ | $\mathbf{0.553}_{\pm.010}$ | $\mathbf{0.638}_{\pm.007}$ | $0.250_{\pm.011}$ | $0.225_{\pm.009}$ | $0.271_{\pm.007}$ |
| OURS | $\mathbf{0.113}_{\pm.000}$ | $\mathbf{0.133}_{\pm.001}$ | $\mathbf{0.218}_{\pm.002}$ | $0.330_{\pm.006}$ | $0.466_{\pm.007}$ | $0.545_{\pm.004}$ | $0.447_{\pm.010}$ | $0.577_{\pm.014}$ | $0.687_{\pm.009}$ | $\mathbf{0.179}_{\pm.011}$ | $\mathbf{0.205}_{\pm.006}$ | $\mathbf{0.257}_{\pm.006}$ |

**Classification (LP)**

| Method | Hypertension (2) | | | | Anxiety (2) | | | | Activity Recognition (20) | | | |
|---|---|---|---|---|---|---|---|---|---|---|---|---|
| | $F_1$ | Acc | BAcc | AUC | $F_1$ | Acc | BAcc | AUC | $F_1$ | Acc | BAcc | AUC |
| OURS + Modality Specific Enc. | $0.686_{\pm.003}$ | $\mathbf{0.695}_{\pm.003}$ | $\mathbf{0.651}_{\pm.003}$ | $\mathbf{0.755}_{\pm.003}$ | $0.552_{\pm.003}$ | $0.536_{\pm.003}$ | $0.535_{\pm.003}$ | $0.744_{\pm.004}$ | $\mathbf{0.490}_{\pm.008}$ | $\mathbf{0.508}_{\pm.009}$ | $\mathbf{0.492}_{\pm.008}$ | $\mathbf{0.902}_{\pm.003}$ |
| OURS | $\mathbf{0.687}_{\pm.003}$ | $0.693_{\pm.004}$ | $\mathbf{0.651}_{\pm.003}$ | $0.754_{\pm.004}$ | $\mathbf{0.690}_{\pm.003}$ | $\mathbf{0.692}_{\pm.004}$ | $\mathbf{0.683}_{\pm.004}$ | $\mathbf{0.758}_{\pm.004}$ | $0.472_{\pm.008}$ | $0.493_{\pm.008}$ | $0.474_{\pm.008}$ | $.899_{\pm.003}$ |

**Regression (LP)**

| Method | Age | | BMI | | Insulin Resis. | |
|---|---|---|---|---|---|---|
| | MAE | Corr | MAE | Corr | MAE | Corr |
| OURS + Modality Specific Enc. | $6.573_{\pm.037}$ | $0.715_{\pm.003}$ | $4.435_{\pm.025}$ | $0.658_{\pm.004}$ | $1.582_{\pm.016}$ | $0.287_{\pm.010}$ |
| OURS | $\mathbf{6.491}_{\pm.035}$ | $\mathbf{0.722}_{\pm.004}$ | $\mathbf{4.383}_{\pm.025}$ | $\mathbf{0.673}_{\pm.004}$ | $\mathbf{1.549}_{\pm.017}$ | $\mathbf{0.321}_{\pm.008}$ |

activity recognition classification tasks, while exhibiting slight degradation for anxiety detection. Finally, modality specific encoding exhibits slight degradation across all regression tasks (age, BMI, insulin resistance). These results are presented in full in Table 12.

It is possible that while modality specific patch encoders enable the model to better learn features unique to each sensor, a single shared projection layer better enables the model to learn shared features and correlations between sensor modalities. Presented with these mixed results we opt for a single shared patch encoder which reduces the number of learnable parameters. Interestingly, we find that modern SOTA long-context wearable sensor foundation models (e.g. LSM, WBM) also use shared modeling for all sensor modalities.

## A.5   ADDITIONAL DISCUSSIONS

### A.5.1   THE UTILITY OF DAY-LEVEL FEATURES

Traditionally, generalist methods for time-series health signals have focused on small windowed segments of data on the order of seconds or sub-seconds (Abbaspourazad et al., 2023; Xu et al., 2024; Narayanswamy et al., 2024b; Yuan et al., 2024). Such methods allow for fine-grain activity and physiological tracking. An adjacent body of work has explored the utility of longer observations, on the order of hours (Spathis et al., 2021; Narayanswamy et al., 2024a), enabling more complex person-level insights. In this work seek to expand the observation window to encode a high-level of context. Day level features allow models to learn relationships not possible from shorter spans, for example, how a person's activity during the day may affect their night-time resting heart rate. Looking forward, we intend to continue exploring how best to encode large context windows to include known week, seasonal, and year level periodicities.

### A.5.2   PERSON-LEVEL VERSUS EVENT-LEVEL PERFORMANCE

Analysis of the discriminative results (classification and regression) presented in the main body of the paper, raise an interesting question: how do generative pre-training affect performance on person-level and event-level tasks. For person-level tasks (hypertension, anxiety, age, BMI) we find that `AIM` consistently outperforms supervised baselines while only using a simple linear probe. In contrast, we find for the event-level task (20-class activity recognition), ResNet50, a supervised baseline performs extremely well, and likely a fully-finetuned `AIM` model is needed to surpass it. This suggest that while supervised methods easily capture event-level features (e.g., sudden heart rate changes due to activity), they struggle to learn slow-changing, near-constant day-level features more-relevant to person-level tasks. This highlights how method, like are own, learn a more complex representation of the data via generative pre-training. We further concede that our contrastive SSL baselines fail to fully realize the gains of pre-training. We hypothesizes that more complex time-series augmentations are needed to leverage their effect.

### A.5.3   LIMITATIONS AND FUTURE WORK

Here we expand upon the limitations and future work introduced in the main body of the paper.

**Generalizing to New Devices.** Though many commodity wearables host a similar suite of sensors there are inevitable differences between these software-hardware systems. We acknowledge that our

methods focuses on a small subset of such devices. Future work will explore the generalizability of our methods to additional devices and datasets, and investigate the extent to which device specific missingness patterns result in a distribution shift.

**Generalizing to Open Data.** Most publicly available wearable datasets (e.g. WESAD (Schmidt et al., 2018), PAMAP2 (Bleser et al., 2015)) are composed of high-frequency raw signals that are very limited in their temporal context with only a subset of the sensors we have available. Thus, they are unable to shown to be used in our setting of day-level context. All of Us (Jeong et al., 2025) demonstrates an interesting avenue to apply our work. Although limited to only the Heart Rate and Step Count channels (compared to our 26 channels), the dataset contains with long context windows and minutely data, and presents an interesting direction in future work to apply our `AIM` method.

**Data and Feature Scales.** Time-series analysis often requires explicit assumptions regarding data scale. As such, our method focuses on day-long samples. We acknowledge that such data disregards known periodicities (e.g., weekly, seasonal, etc.). Future work will explore combining our fine-grained behavioral and physiological modeling with insights from longer windows. Furthermore, our method utilizes minutely aggregated features as opposed to the raw sensor feeds common in sensing research. This is a practical limitation, as data is not stored in its raw form at this scale.

**Handling Sensor Feature.** Our method utilizes 26 features derived from a set of 5 sensors, and regards each feature as independent in the modeling. In reality there are significant correlations between features from the same sensor (e.g., heart rate and heart rate variability). More work can be done to explore how best to combine these multimodal features – potentially sensor-specific encoders, cross-attention, or special class tokens per-sensor feed.

### A.5.4 BROADER IMPACT

Personal and ubiquitous health technologies, including smart phones and wearables, have the potential to scale to billions of individuals. Such devices allow for significant self- and longitudinal tracking, and in so doing may augment the current paradigm of clinical healthcare. To-date, consumer health technologies focus on low-level insights, such as steps, resting heart-rate, and sleep staging, which allow users to reason on personal higher-level insights (e.g., "my resting heart-rate has been elevated ever since I fell sick").

In contrast, our method, trained on day-level samples, learns behavioral and physiological patterns useful in deriving more complex insights. For example, our method shows the potential to predict anxiety and hypertension, insights that humans and commercial algorithms would struggle to derive given only sensor data. We believe this line of work will one day enable people to make the most of their tracked wearable data, better understand their behavior and physiology, and in so doing receive more proactive and better informed care.

