# OpenReview forum: "Self-supervised Learning for Incomplete Multimodal Wearable Sensor Data"
_ICLR.cc/2026/Conference — Submitted to ICLR 2026_

### Official Review · Reviewer_rRLt · 2025-10-30

**Soundness:** 3
**Presentation:** 3
**Contribution:** 3
**Rating:** 6
**Confidence:** 3

**Summary:**

In this study, the authors design a self-supervised pretraining method to pretrain on a large-scale dataset from wearable sensors. The key claim is that most existing studies have not considered the data missing issue. The key contribution of this paper lies in two aspects, the first is the adaptive self-supervised pretraining method, and the second is the large-scale pretraining experiments. For evaluations, the authors design generative, classification and regressive tasks to compare with baseline models.

**Strengths:**

1. This paper targets a practical and important problem in the domain of multimodel wearable data.

2. In experiment, the authors utilize large-scale data for model training. The scale is impressive.

3. The figures are clear and make this paper easier to understand.

**Weaknesses:**

The masking pretraining strategy has been there for a while, as a result, even though this paper claims their adaptive attention masking design in novel, concerns remain for the novelty contribution.

During the model training process, will the training data be complete data (without missing) or data with missing?

In the ablation study, it seems the adaptive masking does not introduce significant improvement.

More detailed explanations of experimental settings. For example, are the data for training and testing come from different subjects? Are the data from watch, smartphone, or other wearable devices?

The current baselines are not the state-of-the art. I would like to suggest the authors to do a thorough literature review. If time permits, please compare with or discuss these more recent SOTA studies: “Crosshar: Generalizing cross-dataset human activity recognition via hierarchical self-supervised pretraining”,  “UniMTS: Unified Pre-training for Motion Time Series”. These work also focus on self-supervised pretrianing in similar application scenarios.

**Questions:**

The masking pretraining strategy has been there for a while, as a result, even though this paper claims their adaptive attention masking design in novel, concerns remain for the novelty contribution.

During the model training process, will the training data be complete data (without missing) or data with missing?

In the ablation study, it seems the adaptive masking does not introduce significant improvement.

More detailed explanations of experimental settings. For example, are the data for training and testing come from different subjects? Are the data from watch, smartphone, or other wearable devices?

The current baselines are not the state-of-the art. I would like to suggest the authors to do a thorough literature review. If time permits, please compare with or discuss these more recent SOTA studies: “Crosshar: Generalizing cross-dataset human activity recognition via hierarchical self-supervised pretraining”,  “UniMTS: Unified Pre-training for Motion Time Series”. These work also focus on self-supervised pretrianing in similar application scenarios.

---

> ### Author Response · Authors · 2025-11-21
> **rRLt Response (1/2)**
>
> Thank you for your review. We appreciate your recognition of the impressive scale of our experiments and the clarity of our figures. We are glad you found the problem of multimodal wearable data to be of high practical importance. To address your questions and improve the paper, we have added:
> * **New Baselines**: We added **mTAN** and **TimesFM** to strengthen our comparisons. We also updated the Related Work to cite and discuss the recent HAR-specific works you suggested, clarifying the domain differences that prevent their direct application here.
> * **Clarified Details**: We revised the text to make participant-level splits and device details (smartwatches/phones) more prominent. Additionally, we confirmed in the text that the model is indeed trained on data with real-world missingness (not complete data), supported by new references to figures.
> * **Clarified Novelty**: We expanded Section 4 to explicitly list the technical contributions of the AIM strategy (joint missingness modeling and mixed masking strategies) to address concerns regarding the novelty of the masking approach.
>
> ---
>
> # 1. Comparing AIM to General Masked Pre-training
> ### Addressing:
> * W1/Q1: “The masking pretraining strategy has been there for a while, as a result, even though this paper claims their adaptive attention masking design in novel, concerns remain for the novelty contribution.”
>
> ### Response:
>
> **The specific novel contributions focused on methodology are as follows:**
> * First to jointly model inherited and artificial missingness for representation learning, enabling the model to recognize missingness as a natural feature of the data.
>
> * First to adapt MAE to allow for variable masking ratios without sacrificing computational efficiency for large-scale pre-training.
>
> * First to propose training with a concurrent mix of masking strategies (i.e. random, temporal bar, modality bar), rather than a single masking strategy (i.e. random masking).
>
> AIM is a simple yet powerful idea that allows us to learn a missingness-aware foundation model. Before the MAE method was published, masked reconstruction was already a widely-studied method in machine learning. MAE established the key details that made it a competitive pre-training method. **Analogously, before AIM, there was no clear understanding of what constituted an effective pre-training strategy for data in the presence of missingness.**
>
> While recently published baselines (e.g. LSM [4] (ICLR ‘25), and WBM [5] (ICML ‘25)) continue to train on imputed data, **AIM demonstrates that the standard imputation approach is not only unnecessary, but is actually suboptimal.** As such, our work addresses a critical, underexplored problem.
>
> # 2. Further Explanations and Details
> ### Addressing
> * W2/Q2 + W4/Q4
>
> ### Response:
> ### W2/Q2: “During the model training process, will the training data be complete data (without missing) or data with missing?”
>
> All of our training and evaluation data (and all of our data in general) is afflicted with missingness. Figure 1 shows an example of our sensor data input, Figure 2 shows the distribution of missingness for all of our datasets combined (note that no samples in our dataset have 0% missingness), and Figure 7 in the Appendix shows a gallery of random data examples with real-world missingness.
>
>
> ### W4/Q4: “More detailed explanations of experimental settings. For example, are the data for training and testing come from different subjects? Are the data from watch, smartphone, or other wearable devices?”
>
> We thank the reviewer for highlighting the need for clarity on these experimental settings. We can confirm these details are in the original manuscript and apologize if they were not sufficiently prominent. We have revised the text to make them more prominent.
> In Line 149, we clarify that participant-level train/test split was used.
> Figure 6 in the Appendix shows the distribution of wearable devices used. Namely, smart watches and the smartphone are used, with the most common devices being Pixel Watch 2, MobileTrack, Fitbit Sense 2, and the Pixel Watch 3.

---

> > ### Author Response · Authors · 2025-11-21
> > **rRLt Response (2/2)**
> >
> > # 3. Benefits of Adaptive Masking
> > ### Addressing:
> > * W3/Q3: “In the ablation study, it seems the adaptive masking does not introduce significant improvement.”
> >
> > ### Response:
> > Thank you for your feedback. To ensure we are aligned on the interpretation of the ablation study, could you help us understand which specific results suggested a lack of improvement? From our perspective, Table 5 demonstrates a significant improvement. First, the "w/o Inherit" ablation in Table 5, performs significantly worse than our full AIM_FM model. This is most evident in the generative tasks, which show a 38-40% error increase (e.g., MSE 0.45 vs. 0.62), but the consistent performance degradation across tasks is also key, as it demonstrates worse generalizability. Second, the "w/o Mixing" ablation, which uses a single fixed-masking strategy, shows a consistent performance drop across almost all tasks, proving it is less generalizable than our model's more robust mask-mixing strategy.
> >
> > # 4. Defining SOTA in Our Long-time-scale Multimodal Setting
> > ### Addressing:
> > * W4/Q4: “The current baselines are not the state-of-the art. I would like to suggest the authors to do a thorough literature review. If time permits, please compare with or discuss these more recent SOTA studies: “Crosshar: Generalizing cross-dataset human activity recognition via hierarchical self-supervised pretraining”, “UniMTS: Unified Pre-training for Motion Time Series”. These work also focus on self-supervised pretrianing in similar application scenarios.”
> >
> > ### Response:
> > Thank you for your feedback. **We would like to clarify that our baselines are state-of-the-art for our setting.**
> > Our data setting is focused on long-context minutely-aggregated multimodal sensor data, and there are two main works in this domain from two different groups that address this setting, LSM [1] and WBM [2]. If we look at the classical high-frequency wearable sensor space, RelCon represents the state-of-the-art for this, demonstrating SOTA representation learning results for both HAR IMU sensor data [3] and PPG sensor data [4].
> >
> > CrossHAR [5] and UniMTS [6] are prominent methods in the wearable domain, but their methods are very specific to high frequency wearable IMU sensors and thus use physical 3D rotations as an augmentation method. These methods are not appropriate for our setting because our sensor data does not have specific x,y,z axis data for 3D rotation. Nonetheless, we appreciate you bringing this to our attention, and **we have cited these papers and added this discussion in the Related Work [Lines 120-121].**
> >
> >
> >
> > # References
> > [1] Scaling Wearable Foundation Models. ICLR 2025
> >
> > [2] Beyond Sensor Data: Foundation Models of Behavioral Data from Wearables Improve Health Predictions. ICML 2025
> >
> > [3] RelCon: Relative Contrastive Learning for a Motion Foundation Model for Wearable Data. ICLR 2025
> >
> > [4] Pulse-PPG: An open-source field-trained PPG foundation model for wearable applications across lab and field settings. ACM IMWUT 2025
> >
> > [5] CrossHAR: Generalizing cross-dataset human activity recognition via hierarchical self-supervised pretraining. ACM IMWUT 2024
> >
> > [6] UniMTS: Unified Pre-training for Motion Time Series. NeurIPS 2024

---

> > > ### Author Response · Authors · 2025-11-26
> > > **Follow-up**
> > >
> > > Dear Reviewer, we wanted to follow-up to see if we have addressed your concerns, or if you have anything else that you would like to further discuss. Thank you so much.

---

> > > > ### Comment · Reviewer_rRLt · 2025-11-27
> > > >
> > > > Thank you for the hard work on the rebuttal, and thank you for the follow-up! Yes, most of my concerns have been addressed. Thanks!

---

### Official Review · Reviewer_9TML · 2025-10-31

**Soundness:** 3
**Presentation:** 4
**Contribution:** 2
**Rating:** 4
**Confidence:** 4

**Summary:**

This paper introduces AIM, a self-supervised framework for large-scale pretraining on incomplete multimodal wearable sensor data, eliminating the need for explicit imputation. The approach combines inherited (real-world) and artificial masks to improve robustness to missingness, leading to AIM_FM, a foundation model trained on 40 million hours of sensor data. The model outperforms existing wearable SSL baselines (e.g., LSM, LIMU-BERT) across generative, classification, and regression tasks and demonstrates strong robustness under targeted missingness scenarios.

**Strengths:**

1. This paper addresses a highly practical and timely challenge—developing large-scale foundation models for time-series wearable data—making it an important direction for real-world deployment.

2. The use of 40 million hours of pretraining data is unprecedented and gives the work a distinctive position in the field.

3. The manuscript is clearly written and easy to follow.

**Weaknesses:**

1. Please discuss additional recent works in multimodal time-series learning that handle missing modalities without explicit imputation (e.g., FlexMoE [1], FuseMoE [2]). Furthermore, techniques such as mTAN [3] and Neural ODEs[4] naturally handle irregular sampling and in-sample missingness, and could provide useful baselines or points of comparison.

2. The algorithmic novelty appears somewhat limited, as the paper mainly engineers existing methods--applying reconstruction loss only on artificial masks---and prior work has proposed native support for missing modalities without imputation. Thus, the methodological contribution feels incremental rather than conceptually new.

3. The scaling laws presented appear closely aligned with those already established in LSM. While the inclusion of a comparison in Figure 4 is helpful, the contribution seems like an extension of prior scaling analyses.

4. For the extrapolation task—which is more similar to time-series forecasting—standard baselines such as Chronos [5], TimesFM, and MoiRai  should be included for a more comprehensive evaluation. Instead of retraining Chronos, the authors could perform zero-shot forecasting to assess how pretrained models generalize under similar conditions as a litmus test. Currently, many of baselines are variants of the proposed approach, making improvements somewhat expected. Broader comparisons would strengthen the paper’s empirical rigor. Additionally, the recently released Chronos-2  supports variable covariates and could be a relevant comparison in future revisions.

5. Is a learnable mask necessary? Representing time series as quantized tokens—similar to Chronos [5]
 or MAESTRO [6]—and reserving distinct symbols for different types of missingness (e.g., type 1 vs. type 2) could be a strong ablation to demonstrate the utility of the learnable mask.

6. Given that one of the paper’s major contributions is the large-scale pretraining dataset and its promised release upon acceptance, this work may be better suited as a dataset/benchmarking paper. Nevertheless, it offers a valuable community resource for researchers lacking access to large-scale wearable data.

References :

[1] FlexMoE, Neurips 2024

[2] FuseMoE, Neurips 2024

[3] mTAN, ICLR 2021

[4] Latent ODEs for Irregularly-Sampled Time Series, Neurips 2019

[5] Chronos, TMLR 2024

[6] MAESTRO, Neurips 2025

**Questions:**

Please see weaknesses.

---

> ### Author Response · Authors · 2025-11-21
> **9TML Response (1/5)**
>
> Thank you for your excellent review and for recognizing the unprecedented scale of our 40-million-hour pre-training dataset. We are glad you found the manuscript clear and the challenge we address to be highly practical for real-world deployment. We have incorporated your suggestions and added the following:
> * **New Baselines**: Per your recommendation, we have added **mTAN** (benchmarking against irregularly sampled methods) and **TimesFM** (zero-shot forecasting), showing AIM_FM outperforms both.
> * **Expanded Related Work**: We added discussions on FlexMoE, FuseMoE, and Neural ODEs to contextualize our work against methods that handle missing modalities without explicit imputation.
> * **Scaling Clarification**: We revised the text to clarify that our scaling results are intended to show that training on imputed data (LSM) leads to premature saturation, rather than claiming scaling itself as the primary novelty.
> * **Tokenization Discussion**: We discussed quantized tokenization (e.g., Chronos, MAESTRO) as an alternative avenue for future research.
>
> # 1. Comparisons to Methods for Irregularly Sampled TS
> ### Addressing:
> * W1: “.. techniques such as mTAN [3] and Neural ODEs[4] naturally handle irregular sampling and in-sample missingness, and could provide useful baselines or points of comparison.”
>
> ### Response:
> **We have added an mTAN baseline  [Tables 2,3,4], and have extended our discussion of irregularly-sampled time-series in “Self-supervised Learning for Other Incomplete Data”  of Section 2 Related Works [Line 149-158].** We previously omitted detailed discussion of irregularly sampled works due to space limitations. We agree that this is a relevant body of work. Our discussion is summarized below and the tables are presented below that:
> Irregularly-sampled time-series represents a similar but different domain from our missingness-prone mobile health setting. Irregularly sampled time-series such as ICU lab measurements [1] are collected at distinct intervals with many other modalities typically missing. In contrast wearables produce regularly-sampled data where modalities drop out in structured groups (Figure 2). The most related work is mTAN [2], which proposes an approach and architecture for learning self-supervised representations directly on multimodal ICU data without imputation. Neural ODEs also learn directly on the irregularly-sampled time-series [3], but have high computational cost and do not necessarily scale to our large-scale pre-training setting [4]. Other related works include HetVAE [5] and Gaussian Process methods [6], which enable explicit uncertainty modeling.
> **We benchmark our AIM_FM against mTAN and show that our model achieves significantly better performance across every task:** generative (i.e. rand imp 80% MSE of  0.218 vs. 0.627), classification (i.e. activity F1 of 0.472 vs. 0.193), and regression (i.e. BMI correlation of 0.673 vs. 0.240).
>
> ---
> #### Generative
> | | Random Imp. (MSE) | | | Temporal Interp. (MSE) | | | Temporal Extrap. (MSE) | | | Signal Imp. (MSE) | | |
> |---|---|---|---|---|---|---|---|---|---|---|---|---|
> | | 30% | 50% | 80% | 10 Min | 30 Min | 60 Min | 10 Min | 30 Min | 60 Min | 2/26 | 6/26 | 12/26 |
> | **Statistical** |
> | Linear Interpolation | 0.572±0.003 | 0.629±0.004 | 0.788±0.006 | 0.523±0.012 | 0.713±0.013 | 0.850±0.013 | 0.749±0.021 | 0.983±0.023 | 1.174±0.026 | - | - | - |
> | Nearest Neighbor | 0.707±0.003 | 0.773±0.004 | 0.952±0.006 | 0.648±0.015 | 0.868±0.014 | 1.030±0.014 | 0.749±0.020 | 0.983±0.023 | 1.174±0.026 | - | - | - |
> | Mean | 0.924±0.005 | 0.957±0.005 | 0.949±0.006 | 0.929±0.015 | 0.932±0.016 | 0.963±0.014 | 1.096±0.022 | 1.096±0.018 | 1.086±0.016 | 1.238±0.025 | 1.258±0.015 | 1.268±0.015 |
> | **Deep Learning** | | | | | | | | | | | | |
> | [NEW] TimesFM | - | - | - | - | - | - | 0.548±0.051 | 0.687±0.060 | 0.857±0.059 | - | - | - |
> | Limu-bert | - | - | - | 1.010±0.017 | 1.062±0.016 | 1.062±0.016 | 1.154±0.021 | 1.159±0.020 | 1.159±0.015 | - | - | - |
> | [NEW] mTAN | 0.605±0.002 | 0.617±0.002 | 0.627±0.003 | 0.647±0.016 | 0.697±0.011 | 0.731±0.010 | 0.721±0.014 | 0.803±0.019 | 0.929±0.022 | 0.741±0.011 | 0.790±0.007 | 0.958±0.010 |
> | LSM | 0.146±0.000 | 0.178±0.001 | 0.293±0.001 | 0.605±0.010 | 0.687±0.008 | 0.717±0.011 | 0.670±0.020 | 0.746±0.012 | 0.775±0.009 | 0.683±0.041 | 0.561±0.016 | 0.443±0.005 |
> | **Ours** | **0.113±0.000** | **0.133±0.001** | **0.218±0.002** | **0.330±0.006** | **0.466±0.007** | **0.545±0.004** | **0.447±0.010** | **0.577±0.014** | **0.687±0.009** | **0.179±0.011** | **0.205±0.006** | **0.257±0.006** |

---

> > ### Author Response · Authors · 2025-11-21
> > **9TML Response (2/5)**
> >
> > #### Classification
> > || Hypertension (2) |||| Anxiety (2) |||| Activity (20) ||||
> > |---|---|---|---|---|---|---|---|---|---|---|---|---|
> > || F1 | Acc | BAcc | AUC | F1 | Acc | BAcc | AUC | F1 | Acc | B Acc | AUC |
> > | **Fully-Supervised** |
> > | ResNet | 0.529±0.003 | 0.587±0.003 | 0.516±0.003 | 0.624±0.004 | 0.655±0.003 | 0.651±0.003 | 0.645±0.003 | 0.709±0.003 | 0.721±0.007 | 0.734±0.007 | 0.729±0.007 | 0.965±0.002 |
> > | ViT-1D | 0.516±0.003 | 0.509±0.004 | 0.481±0.003 | 0.520±0.005 | 0.597±0.004 | 0.586±0.004 | 0.583±0.004 | 0.620±0.004 | 0.367±0.008 | 0.374±0.008 | 0.351±0.008 | 0.863±0.004 |
> > | [NEW] **Ours (FT)** | 0.680±0.003 | 0.648±0.004 | 0.626±0.003 | 0.703±0.004 | **0.693±0.003** | **0.693±0.003** | **0.685±0.003** | **0.759±0.003** | **0.767±0.007** | **0.779±0.007** | **0.774±0.007** | **0.978±0.001** |
> > | **Frozen + Linear Probe** |
> > | Limu-bert | 0.599±0.003 | 0.596±0.004 | 0.561±0.003 | 0.635±0.004 | 0.640±0.003 | 0.641±0.003 | 0.632±0.003 | 0.693±0.004 | 0.190±0.006 | 0.219±0.008 | 0.191±0.006 | 0.735±0.004 |
> > | WBM | 0.582±0.004 | 0.572±0.004 | 0.542±0.004 | 0.599±0.004 | 0.605±0.003 | 0.604±0.003 | 0.597±0.003 | 0.643±0.004 | 0.107±0.005 | 0.117±0.006 | 0.102±0.005 | 0.611±0.006 |
> > | RelCon | 0.564±0.004 | 0.565±0.005 | 0.530±0.005 | 0.590±0.006 | 0.615±0.004 | 0.609±0.004 | 0.604±0.004 | 0.652±0.005 | 0.058±0.004 | 0.050±0.000 | 0.005±0.000 | 0.509±0.005 |
> > |[NEW] mTAN | 0.637±0.003 | 0.576±0.004 | 0.566±0.003 | 0.607±0.005 | 0.506±0.003 | 0.582±0.003 | 0.486±0.003 | 0.674±0.003 | 0.193±0.006 | 0.217±0.007 | 0.192±0.006 | 0.731±0.004 |
> > | SimCLR | 0.524±0.003 | 0.548±0.004 | 0.501±0.003 | 0.568±0.004 | 0.603±0.003 | 0.601±0.003 | 0.594±0.003 | 0.636±0.003 | 0.109±0.005 | 0.124±0.007 | 0.098±0.005 | 0.652±0.005 |
> > | Dino | 0.536±0.004 | 0.504±0.004 | 0.487±0.004 | 0.510±0.004 | 0.557±0.003 | 0.562±0.003 | 0.551±0.003 | 0.582±0.004 | 0.110±0.005 | 0.124±0.007 | 0.102±0.005 | 0.635±0.005 |
> > | MSN | 0.555±0.003 | 0.552±0.004 | 0.519±0.003 | 0.575±0.004 | 0.547±0.004 | 0.551±0.004 | 0.515±0.003 | 0.571±0.004 | 0.144±0.006 | 0.159±0.008 | 0.136±0.006 | 0.692±0.005 |
> > | LSM | 0.676±0.003 | 0.682±0.004 | 0.640±0.003 | 0.739±0.004 | 0.678±0.003 | 0.678±0.003 | 0.670±0.003 | 0.743±0.004 | 0.470±0.008 | 0.489±0.008 | 0.470±0.008 | **0.900±0.003** |
> > | **Ours** | **0.687±0.003** | **0.693±0.004** | **0.651±0.003** | **0.754±0.004** | **0.690±0.003** | **0.692±0.004** | **0.683±0.004** | **0.758±0.004** | **0.472±0.008** | **0.493±0.008** | **0.474±0.008** | 0.899±0.003 |
> >
> > #### Regression
> > | | Age | | BMI | | IR | |
> > |---|---|---|---|---|---|---|
> > | | MAE | Corr | MAE | Corr | MAE | Corr |
> > | **Fully-Supervised** |
> > | ResNet | 7.429±0.039 | 0.618±0.004 | 5.067±0.028 | 0.515±0.005 | 1.640±0.018 | 0.241±0.011 |
> > | ViT-1D | 9.653±0.049 | 0.132±0.006 | 6.061±0.035 | 0.047±0.006 | 1.580±0.016 | 0.139±0.009 |
> > | [NEW] Ours (FT) | 7.574±0.037 | 0.606±0.004 | 5.172±0.032 | 0.522±0.007 | **1.435±0.020** | 0.291±0.009 |
> > | **Frozen + Linear Probe** |
> > | Limu-bert | 8.445±0.038 | 0.475±0.005 | 5.486±0.029 | 0.408±0.004 | 1.599±0.017 | 0.223±0.009 |
> > | WBM | 8.614±0.036 | 0.449±0.005 | 5.662±0.029 | 0.352±0.006 | 1.714±0.017 | 0.227±0.010 |
> > | RelCon | 8.886±0.044 | 0.380±0.006 | 5.527±0.031 | 0.407±0.007 | 1.611±0.017 | 0.189±0.009 |
> > | [NEW] mTAN | 9.088±0.046 | 0.334±0.008 | 5.835±0.035 | 0.240±0.007 | 1.577±0.017 | 0.204±0.010 |
> > | SimCLR | 9.207±0.042 | 0.345±0.005 | 5.852±0.028 | 0.235±0.005 | **1.546±0.016** | 0.248±0.009 |
> > | Dino | 9.685±0.044 | 0.112±0.006 | 5.968±0.036 | 0.122±0.008 | 1.588±0.016 | 0.087±0.011 |
> > | MSN | 9.416±0.042 | 0.255±0.006 | 5.837±0.036 | 0.250±0.006 | 1.573±0.016 | 0.236±0.010 |
> > | LSM | **6.409±0.033** | **0.728±0.003** | 4.390±0.027 | 0.667±0.004 | 1.595±0.017 | 0.304±0.008 |
> > | Ours | 6.491±0.035 | 0.722±0.004 | **4.383±0.025** | **0.673±0.004** | 1.549±0.017 | **0.321±0.008** |
> >
> > ---

---

> > > ### Author Response · Authors · 2025-11-21
> > > **9TML Response (3/5)**
> > >
> > > # 2. Our Missingness Modes Subsumes Modality Missingness for Self-Supervised Learning
> > > ### Addressing:
> > > * W1: “ Please discuss additional recent works in multimodal time-series learning that handle missing modalities without explicit imputation (e.g., FlexMoE [1], FuseMoE [2]) ...”
> > > * W2: “ … prior work has proposed native support for missing modalities without imputation.”
> > >
> > > ### Response:
> > > Thank you for the suggestion. We agree these related works assist the reader in contextualizing our contributions. We now explicitly discuss prior works on missing modality handling in “Supervised Learning for Incomplete Data” in Section 2 Related Work [Line 165-172].** Due to space restrictions, we could not include this within our initial submission, but we have added this into the current version. In summary, AIM extends such works and differentiates itself from them in the following ways:
> > > Missing modalities are a subset of the missingness patterns encountered in our mobile health data. In the missing modality setting, a modality will be missing across the entire time-series, but in our setting, specific modalities will drop in and out over time due to a variety of factors, such as packet loss, strategic de-activation for battery conservation, and intermittent loosening. Please see Figure 2 for a visualization of our missingness modes.
> > > This temporal component is not addressed by Flex-MoE [10], which assumes if the modality is present, then it is fully observed. Fuse-MoE [9] does handle missingness over time via mTAN [2], but our new experiments with mTAN demonstrate that AIM is stronger across every task. This is also why, in our “Robustness to Targeted Missingness” experiment in Figure 5, we not only demonstrate robustness amongst dropped modalities, but also dropped temporal windows (i.e. removing all nighttime data).
> > > Additionally, we note that Flex-MoE and Fuse-MoE are designed for supervised learning, and it is unclear how to extend them for self-supervised learning. AIM is able to unify its ability to handle missingness with self-supervised learning, demonstrating a simple yet powerful approach for scalable pre-training.
> > >
> > >
> > > # 3. Clarifying Claims with Respect to the Method
> > > ### Addressing:
> > > * W2: “The algorithmic novelty appears somewhat limited, as the paper mainly engineers existing methods--applying reconstruction loss only on artificial masks..”
> > >
> > > ### Response:
> > > Thank you for giving us the opportunity to expand on this, we have added this discussion in Section 4 AIM Methodology [Lines 212-247]. **The specific novel algorithmic contributions are as follows:**
> > > * First to jointly model inherited and artificial missingness for representation learning, enabling the model to recognize missingness as a natural feature of the data.
> > >
> > > * First to adapt MAE to allow for variable masking ratios without sacrificing computational efficiency for large-scale pre-training.
> > >
> > > * First to propose training with a concurrent mix of masking strategies (i.e. random, temporal bar, modality bar), rather than a single masking strategy (i.e. random masking).
> > >
> > > AIM is a simple yet powerful idea that allows us to learn a missingness-aware foundation model. Before the MAE method was published, masked reconstruction was already a widely-studied method in machine learning. MAE established the key details that made it a competitive pre-training method. **Analogously, before AIM, there was no clear understanding of what constituted an effective pre-training strategy for data in the presence of missingness.**
> > >
> > > While recently published baselines (e.g. LSM [4] (ICLR ‘25), and WBM [5] (ICML ‘25)) continue to train on imputed data, **AIM demonstrates that the standard imputation approach is not only unnecessary, but is actually suboptimal.** As such, our work addresses a critical, underexplored problem.

---

> ### Author Response · Authors · 2025-11-21
> **9TML Response (4/5)**
>
> # 5. Comparison to the TimesFM forecasting models and Baseline Clarification.
> ### Addressing:
> * W4: “For the extrapolation task—which is more similar to time-series forecasting—... TimesFM … should be included for a more comprehensive evaluation. Instead of retraining … could perform zero-shot forecasting … Currently, many baselines are variants of the proposed approach, making improvements somewhat expected. ”
>
> ### Response:
>
> Thank you for the great suggestion. **We have added TimesFM 2.0 with 500m parameters as a zero-shot forecasting baseline** for our temporal extrapolation task. This can be seen in the “Generative Table” in Rebuttal Section 1, and in our paper’s Table 3 [Lines 372-381]. Our results demonstrate that although TimesFM is able to achieve the second strongest performance out of our baselines, **AIM_FM model still performs the best at the temporal extrapolation task, while also being able to do Random Imputation, Temporal Interpolation, and Signal Imputation**.
>
> Additionally, we'd like to respectfully clarify our perspective on the baseline comparison. We acknowledge that LSM is architecturally similar to our approach. However, we wish to clarify that our other original baselines (e.g., WBM, RelCon, LimuBERT) were already selected to represent distinctively different paradigms from our own, and your suggestions to add mTAN and TimesFM were excellent for further improving diversity. All other deep learning baselines represent distinctively different paradigms across types of data, learning objectives, and whether they handle missingness. Please see the table below for further comparison.
>
> | Baseline | Original Data Type | Learning Objective | Handles Missingness |
> |---|---|---|---|
> | Our AIM_FM | Aggregated Multimodal | Masked across Patches | Yes |
> | LSM | Aggregated Multimodal | Masked across Patches | No |
> | WBM | Aggregated Multimodal | Contrastive with Subjects| No |
> | RelCon | High Frequency | Contrastive with Motifs | No |
> | LimuBERT | High Frequency | Masked across Modalities | No |
> | SimCLR/Dino/MSN | Any | Contrastive with Augmentations| No |
> | ViT1D/ResNet | Any | Supervised | No |
> | [NEW] mTAN | Irregularly Sampled | Masked across Time Points | Yes |
> | [NEW] TimesFM | Any | Forecasting | No |
>
>
> # 6. Further exploring a quantized tokenization approach
> ### Addressing:
> * W4: “Is a learnable mask necessary? Representing time series as quantized tokens—similar to Chronos [5] or MAESTRO [6]—and reserving distinct symbols for different types of missingness (e.g., type 1 vs. type 2) could be a strong ablation to demonstrate the utility of the learnable mask.”
>
> ### Response:
> This is an interesting proposed approach. We acknowledge that these deterministic quantized tokenization approaches offer an intriguing alternative approach to our learned convolutional patch tokenization. Chronos [14] demonstrates this in a forecasting context and MAESTRO [15] demonstrates this in a supervised learning context. However, it is still unclear how they can be used for self-supervised representation learning. For self-supervised learning, TOTEM [16], Haresamudram et al. [17], Pradeepkumar et. al. [18],  instead use learned quantized tokenization approaches. This learned tokenization is more similar to our approach, but these methods use a 2 step approach to first train the VQ-VAE tokenizer then freeze the tokenizer to train their foundation models, whereas our method learns our foundation model with the tokenization together in 1 step.
>
> Therefore, representing the time-series as quantized tokens represents a fundamental architectural and paradigm shift from the core premise of our work, which seeks to unify the MAE large-scale pre-training framework for incomplete sensor data in a simple yet powerful approach. Exploring tokenization is an exciting future research direction that remains quite nascent (e.g. MAESTRO [15], the most related work, is a concurrent paper that has not yet been officially published in the NeurIPS 2025 proceedings).

---

> > ### Author Response · Authors · 2025-11-21
> > **9TML Response (5/5)**
> >
> > # 7. Major Contribution
> > ### Addressing:
> > * W5: “Given that one of the paper’s major contributions is the large-scale pretraining dataset and its promised release upon acceptance, this work may be better suited as a dataset/benchmarking paper. Nevertheless, it offers a valuable community resource for researchers lacking access to large-scale wearable data.”
> >
> > ### Response:
> > We appreciate your acknowledgement of our contribution to an open-source release of our data and code. We would like to acknowledge that ICLR does *not* have an explicit dataset/benchmarking track. However, **in the call for papers (https://iclr.cc/Conferences/2026/CallForPapers), ICLR explicitly asks for “datasets and benchmarks”**, so we submitted this to the main track.
> >
> > # References
> >
> > [1] Predicting in-hospital mortality of ICU patients: The PhysioNet/Computing in Cardiology Challenge 2012. IEEE CinC 2012
> >
> > [2] Multi-time attention networks for irregularly sampled time series. ICLR 2021
> >
> > [3] Latent ordinary differential equations for irregularly-sampled time series. NeurIPS 2019
> >
> > [4] How to train your neural ODE: the world of Jacobian and kinetic regularization. ICML 2020
> >
> > [5] Heteroscedastic temporal variational autoencoder for irregularly sampled time series. ICLR 2022
> >
> > [6] A scalable end-to-end Gaussian process adapter for irregularly sampled time series classification. NeurIPS 2016
> >
> > [7] Supervised learning from incomplete data via an EM approach. NeurIPS 1993
> >
> > [8] How to deal with missing data in supervised deep learning? ICLR 2022
> >
> > [9] FuseMoE: Mixture-of-experts transformers for fleximodal fusion. NeurIPS 2024
> >
> > [10] Flex-MoE: Modeling arbitrary modality combination via the flexible mixture-of-experts. NeurIPS 2024
> >
> > [11] Beyond Sensor Data: Foundation Models of Behavioral Data from Wearables Improve Health Predictions. ICML 2025
> >
> > [12] Limu-bert: Unleashing the potential of unlabeled data for IMU sensing applications. SenSys 2021
> >
> > [13] Scaling Wearable Foundation Models. ICLR 2025
> >
> > [14] Chronos: Learning the language of time series. arXiv 2024
> >
> > [15] MAESTRO: Adaptive Sparse Attention and Robust Learning for Multimodal Dynamic Time Series. NeurIPS 2025
> >
> > [16] Totem: Tokenized time series embeddings for general time series analysis. TMLR 2024
> >
> > [17] Towards learning discrete representations via self-supervision for wearables-based human activity recognition. Sensors 2024
> >
> > [18] Single-channel EEG tokenization through time-frequency modeling. NeurIPS Time-Series 4 Health 2025

---

> > > ### Author Response · Authors · 2025-11-26
> > > **Follow-up**
> > >
> > > Dear Reviewer, we wanted to follow-up to see if we have addressed your concerns, or if you have anything else that you would like to further discuss. Thank you so much.

---

### Official Review · Reviewer_jrDJ · 2025-10-31

**Soundness:** 2
**Presentation:** 3
**Contribution:** 2
**Rating:** 2
**Confidence:** 4

**Summary:**

The paper proposes a self-supervised learning framework for incomplete multimodal wearable sensor data, introducing a method named AIM that extends masked autoencoding to handle missing signals without explicit imputation. The method combines inherited and artificial masking strategies to model missingness patterns directly during pre-training. The approach is evaluated on a multimodal health dataset with generative and predictive downstream tasks.

**Strengths:**

+ The problem is relevant and timely, given the prevalence of incomplete sensor data in wearable and healthcare contexts.
+ The paper provides a systematic experimental setup with reconstruction and regression/classification tasks to validate model performance.
+ The integration of generative and predictive evaluation is methodologically sound, providing a holistic assessment of the learned representations.

**Weaknesses:**

- The proposed method primarily combines existing ideas from MAE-based reconstruction and masking strategies (inherited/artificial masking). While practically useful, it does not introduce a fundamentally new self-supervised objective or architecture.
-  The evaluation relies heavily on a single dataset. To claim generalizability across sensor modalities, additional experiments on multiple benchmark HAR datasets would be necessary.
- Performance gains over baselines are moderate and may not justify publication at a top venue without broader validation or deeper theoretical insights.
-  The paper does not sufficiently compare with the most recent SSL methods for multimodal HAR. Such comparisons are crucial for situating the contribution within the current state of the art.

**Questions:**

1. How generalizable is AIM across domains or datasets with different missingness structures?
2. Would combining AIM with cross-modal alignment objectives further improve robustness?
3. Can the authors justify why their adaptation of MAE is preferable over recent SSL methods explicitly designed for multimodal sensor data?

---

> ### Author Response · Authors · 2025-11-21
> **jrDJ Response (1/5)**
>
> Thank you for your review and for acknowledging that our problem is timely and relevant. We also appreciate your validation of our systematic experimental setup and the soundness of integrating generative and predictive evaluations. Based on your feedback, we have added the following to the paper:
> * **Our Setting vs. HAR Distinction**: We added a section to clarify how our long-context, aggregate setting differs from traditional high-frequency HAR, explaining why specific HAR dataset/baselines are not applicable.
> * **New Baselines**: We added **mTAN** (from irregularly sampled time series) and **TimesFM** (for forecasting) to better situate our work within the literature.
> * **Modality-Specific Ablation**: To address cross-modal alignment, we ran a new experiment implementing modality-specific encoders, which validated that our shared encoder approach yields better generalizability.
> * **Clarified Novelty**: We expanded the methodology section to explicitly detail the unique contributions of AIM regarding variable masking ratios and mixed masking strategies.
>
> ---
>
> # 1. Our Setting Differs Significantly from the Traditional HAR Setting
> ### Addressing:
> * W2: The evaluation relies heavily on a single dataset. To claim generalizability across sensor modalities, additional experiments on multiple benchmark HAR datasets would be necessary.
> * W4: “The paper does not sufficiently compare with the most recent SSL methods for multimodal HAR. Such comparisons are crucial for situating the contribution within the current state of the art.”
>
> ### Response:
> Thank you for this point, and we have now added a dedicated section on this discussion in the Section 2 Related Work [Lines 111-133] to help ensure clarity.
>
> We would like to make it clear that **our setting differs significantly from the HAR field**. Please see the comparison table below. As a consequence, datasets and baselines derived from the traditional HAR domain are not directly applicable to the long-context, aggregate setting we explore here.
>
> |  | Resolution | Context | Modalities | Eval Tasks |
> |---|---|---|---|---|
> | Our Setting | Aggregate (e.g. Minutely) | Long-context (e.g. Day) | IMU, PPG, EDA, Temp, Altimeter | Physiological (Hypertension, IR), Mental (Anxiety), Motion (Activity), Demographic (Age, BMI) |
> | HAR Setting | High Freq (e.g. ~100 Hz) | Short-context (e.g. Seconds) | IMU | Motion (Activity, Gait) |
>
> **Activity classification is included as one of our many evaluation tasks, but it is defined very differently from the traditional HAR domain.** In our setting, an activity is retroactively predicted from an entire day’s worth of data. In the traditional HAR setting, an activity is on-the-fly predicted from a few seconds of data. Next we elaborate on the specific details.
>
> ### Comparison to other/HAR datasets.
> **There are no public datasets that match our setting of day-long multimodal wearable sensor data**. As discussed throughout the paper, in order to help address this, and as adamant supporters of open science, we will release our metabolic study dataset for the community to use.
>
> **All publicly available wearable datasets either have short time windows (e.g. seconds) and/or only contain a subset of our sensing modalities.** A prominent HAR dataset, PAMAP2 [12] only utilizes accelerometer and gyroscope sensors with 10 seconds of data per label. More broadly, WESAD [15] is a wearable dataset that contains a variety of sensing modalities (i.e. PPG, Accelerometer, EDA, and temperature), but it is a lab study with 1 minute of data per label. All of Us [16] does have real-world day-long sensor data, but is limited to only the Heart Rate and Step channels (compared to our 26 total channels).
>
>
> ### Comparison to  other/HAR Baselines.
> We believe that our baselines are state-of-the-art. **There are only 2 methods that have focused on long-context multimodal wearable sensors, both of which we benchmark against**, LSM [4] (ICLR ‘25) and WBM [5] (ICML ‘25). For completeness, **we do evaluate against recent methods originally proposed for HAR and short-context sensor data** such as RelCon [8] (ICLR ‘25) and Limu-bert [6]. AIM_FM performs better than all of these baselines.
>
> **Other recent HAR methods**, such as CrossHAR [13] and UniMTS [14] use physical 3D rotations as an augmentation method. As such they **cannot be run on our data which does not have specific x,y,z axis data.**
>
> We do, however, agree that additional SSL baselines may be beneficial. We have added two baselines in this rebuttal. mTAN [10] (for all tasks) - a method that helps position our work against methods for irregularly sampled multimodal time-series, and TimesFM [11] (for forecasting tasks) which better contextualizes the utility of pre-trained generalist time-series models when applied to wearable sensor data. These updated results may be found in our updated paper and below:

---

> ### Author Response · Authors · 2025-11-21
> **jrDJ Response (2/5)**
>
> # 2. Methodological Novelty Claims
> ### Addressing:
> * W1: The proposed method primarily combines existing ideas from MAE-based reconstruction and masking strategies (inherited/artificial masking). While practically useful, it does not introduce a fundamentally new self-supervised objective or architecture.
>
> ### Response:
> Thank you for giving us the opportunity to expand on this, we have added this discussion in Section 4 AIM Methodology [Lines 212-247]. **The specific novel methodological contributions are as follows:**
>
> * First to jointly model inherited and artificial missingness for representation learning, enabling the model to recognize missingness as a natural feature of the data.
>
> * First to adapt MAE to allow for variable masking ratios without sacrificing computational efficiency for large-scale pre-training.
>
> * First to propose training with a concurrent mix of masking strategies (i.e. random, temporal bar, modality bar), rather than a single masking strategy (i.e. random masking).
>
> AIM is a simple yet powerful idea that allows us to learn a missingness-aware foundation model. Before the MAE method was published, masked reconstruction was already a widely-studied method in machine learning. MAE established the key details that made it a competitive pre-training method. **Analogously, before AIM, there was no clear understanding of what constituted an effective pre-training strategy for data in the presence of missingness.**
>
> While recently published baselines (e.g. LSM [4] (ICLR ‘25), and WBM [5] (ICML ‘25)) continue to train on imputed data, **AIM demonstrates that the standard imputation approach is not only unnecessary, but is actually suboptimal.** As such, our work addresses a critical, underexplored problem.

---

> > ### Author Response · Authors · 2025-11-21
> > **jrDJ Response (3/5)**
> >
> > # 3. Consistent Performance Gains over the Baselines, especially in Generative Tasks
> > ### Addressing:
> > * W3: Performance gains over baselines are moderate and may not justify publication at a top venue without broader validation or deeper theoretical insights.
> >
> > ### Response:
> > Thank you for allowing us to further discuss this. **We highlight that AIM_FM achieves the top performance on 86.7% of the evaluated metrics, outperforming a comprehensive baseline suite across a diverse range of tasks.** Our 95% confidence intervals confirm that these gains are statistically significant. Strong consistency is the key assessment metric, as the purpose of a foundation model is to learn a representation useful for a diverse set of tasks. Additionally, it is worth noting that **AIM_FM achieves a significant 35.6% improvement across the 12 generative tasks compared against LSM, the most closely related work, as well as a substantial 22.1% improvement against TimesFM, our newly added forecasting foundation model baseline.**
> >
> > Furthermore, we would like to emphasize that we have identified deep insights in this field, by demonstrating how **the standard practice of imputation for long-context wearable data is not only unnecessary, but is actually suboptimal.** By natively modeling missingness, we show that,
> >
> > 1) AIM_FM achieves stronger and more consistent downstream generalizability (described above),
> >
> > 2) AIM_FM achieves improved scaling (Figure 4) suggesting that training on imputed data may cause premature saturation.
> >
> > 3) AIM_FM is more robust to targeted missingness, while reflecting clinical domain knowledge (Figure 5 shows how nighttime data is useful for hypertension prediction, reflecting existing clinical knowledge on nighttime resting heart rate).
> >
> > 4) Missingness cannot be simply solved with “stronger” imputation methods, as “stronger” imputation methods, counterintuitively, may lead to worse downstream performance (Table 11).
> >
> >
> > **We assert that we have made a substantial contribution to our field by calling to attention the importance of “respecting” missingness, a natural and ubiquitous artifact of time-series and wearable sensor data.** If this is not sufficient, can you please clarify in what way you believe the performance gains to be moderate?

---

> > > ### Author Response · Authors · 2025-11-21
> > > **jrDJ Response (4/5)**
> > >
> > > # 4. Addressing Specific Questions
> > > ### Addressing:
> > > * Q1, Q2, Q3
> > >
> > > ### Response:
> > > ### Q1: How generalizable is AIM across domains or datasets with different missingness structures?
> > > AIM is generalizable across domains with differing missingness structures as **it does not assume specific missingness patterns, but rather that there exists some underlying missingness.** Even in domains without missingness, our proposed masking mix allows for a more diverse and challenging pre-training task. This was originally discussed in Lines 522-528.
> > >
> > > Our targeted missingness experiment (Figure 5), demonstrates that AIM is robust to different, diverse missingness structures. Furthermore, Appendix A.1.2. shows how our model was pre-trained on ~20 different devices, demonstrating broad generalizability across varying sensor technologies and differing inherent missingness patterns.
> > >
> > >
> > > ### Q2:  Would combining AIM with cross-modal alignment objectives further improve robustness?
> > > Our model is already able to learn cross-modal alignment via the transformer encoder’s attention mechanism, but we agree that adding an additional  cross-modal alignment, similar to the approaches used in FlexMoE [17] and FuseMoE [18], may be useful for further improving robustness. However, these two methods and many related works are designed for supervised learning, and it is unclear how to extend them for self-supervised learning. SleepFM [19] offers an alternative self-supervised approach via CLIP-style cross-modal alignment, but fails to explicitly capture within-modality information and uses a simple fusion approach via concatenation.
> > >
> > > In order to investigate whether a more modality-specific architecture would improve performance, we run a new experiment to modify our shared patch tokenizer into be modality-specific. This can be found in the Section 6 Results [Line 466-475] with details in the Appendix A.4.5 and Table 11. In summary, the new results confirm that this sensor-specific adaption does not add significant value, suggesting that AIM_FM already learns a powerful cross-modal representation via shared encoder layers. **Learning cross-modal alignment is an open research question that we will explore in future work.**
> > >
> > > ---
> > > #### Generative
> > > |  | Random Imp. (MSE)  |  |  | Temporal Interp. (MSE)  |  |  | Temporal Extrap. (MSE)  |  |  | Signal Imp. (MSE)  |  |  |
> > > |---|---|---|---|---|---|---|---|---|---|---|---|---|
> > > |  | 30% | 50% | 80% | 10 Min | 30 Min | 60 Min | 10 Min | 30 Min | 60 Min | 2/26 | 6/26 | 12/26 |
> > > | **AIM_FM** | **0.113±0.000** | **0.133±0.001** | **0.218±0.002** | 0.330±0.006 | 0.466±0.007 | 0.545±0.004 | 0.447±0.010 | 0.577±0.014 | 0.687±0.009 | **0.179±0.011** | **0.205±0.006** | **0.257±0.006** |
> > > | **Modality-specific AIM_FM** | 0.119±0.000 | 0.138±0.001 | 0.223±0.002 | **0.323±0.006** | **0.446±0.007** | **0.533±0.007** | **0.436±0.013** | **0.553±0.010** | **0.638±0.007** | 0.250±0.011 | 0.225±0.009 | 0.271±0.007 |
> > >
> > > #### Classification
> > > |  | Hypertension (2) |  |  |  | Anxiety (2) |  |  |  | Activity (20) |  |  |  |
> > > |---|---|---|---|---|---|---|---|---|---|---|---|---|
> > > |  | F1 | Acc | BAcc | AUC | F1 | Acc | BAcc | AUC | F1 | Acc | B Acc | AUC |
> > > | **AIM_FM** | **0.687±0.003** | 0.693±0.004 | 0.651±0.003 | 0.754±0.004 | **0.690±0.003** | **0.692±0.004** | **0.683±0.004** | **0.758±0.004** | 0.472±0.008 | 0.493±0.008 | 0.474±0.008 | 0.899±0.003 |
> > > | **Modality-specific AIM_FM** | 0.686±0.003 | **0.695±0.003** | 0.651±0.003 | **0.755±0.003** | 0.552±0.003 | 0.536±0.003 | 0.535±0.003 | 0.744±0.004 | **0.490±0.008** | **0.508±0.009** | **0.492±0.008** | **0.902±0.003** |
> > >
> > > #### Regression
> > > |  | Age |  | BMI |  | IR |  |
> > > |---|---|---|---|---|---|---|
> > > |  | MAE | Corr | MAE | Corr | MAE | Corr |
> > > | **AIM_FM** | **6.491±0.035** | **0.722±0.004** | **4.383±0.025** | **0.673±0.004** | **1.549±0.017** | **0.321±0.008** |
> > > | **Modality-specific AIM_FM** | 6.573±0.037 | 0.715±0.003 | 4.435±0.025 | 0.658±0.004 | 1.582±0.016 | 0.287±0.010 |
> > >
> > > ---
> > >
> > >
> > > ### Q3: Can the authors justify why their adaptation of MAE is preferable over recent SSL methods explicitly designed for multimodal sensor data?
> > >
> > > As previously discussed, in Rebuttal Section 1, **most of the prior SSL methods designed for multimodal sensor data are not applicable** to our setting. If you have additional suggestions on baselines, we can work on adding them before the rebuttal period ends.

---

> > > > ### Author Response · Authors · 2025-11-21
> > > > **jrDJ Response (5/5)**
> > > >
> > > > # References
> > > > [1] Masked Autoencoders Are Scalable Vision Learners. CVPR 2022
> > > >
> > > > [2] Generative Pretraining From Pixels. ICML 2020
> > > >
> > > > [3] An Image is Worth 16x16 Words: Transformers for Image Recognition at Scale. ICLR 2021
> > > >
> > > > [4] Scaling Wearable Foundation Models. ICLR 2025
> > > >
> > > > [5] Beyond Sensor Data: Foundation Models of Behavioral Data from Wearables Improve Health Predictions. ICML 2025
> > > >
> > > > [6] Limu-bert: Unleashing the potential of unlabeled data for imu sensing applications. SenSys 2021
> > > >
> > > > [7] Predicting mortality of ICU patients using a cascaded SVM-GLM paradigm. IEEE CinC 2012
> > > >
> > > > [8] RelCon: Relative Contrastive Learning for a Motion Foundation Model for Wearable Data. ICLR 2025
> > > >
> > > > [9] Masked autoencoders that listen. NeurIPS 2022
> > > >
> > > > [10] Multi-time attention networks for irregularly sampled time series. ICLR 2021
> > > >
> > > > [11] A decoder-only foundation model for time-series forecasting. ICML 2024
> > > >
> > > > [12] Introducing a New Benchmarked Dataset for Activity Monitoring. IEEE ISWC 2012
> > > >
> > > > [13] CrossHAR: Generalizing cross-dataset human activity recognition via hierarchical self-supervised pretraining. ACM IMWUT 2024
> > > >
> > > > [14] UniMTS: Unified Pre-training for Motion Time Series. NeurIPS 2024
> > > >
> > > > [15] Introducing WESAD, a multimodal dataset for wearable stress and affect detection. ACM ICMI 2018
> > > >
> > > > [16] The “All of Us” research program. NEJM 2019
> > > >
> > > > [17] Flex-MoE: Modeling arbitrary modality combination via the flexible mixture-of-experts. NeurIPS 2024
> > > >
> > > > [18] FuseMoE: Mixture-of-experts transformers for fleximodal fusion. NeurIPS 2024
> > > >
> > > > [19] SleepFM: Multi-modal representation learning for sleep across brain activity, ECG and respiratory signals. ICML 2024

---

> > > > > ### Author Response · Authors · 2025-11-26
> > > > > **Follow-up**
> > > > >
> > > > > Dear Reviewer, we wanted to follow-up to see if we have addressed your concerns, or if you have anything else that you would like to further discuss. Thank you so much.

---

### Official Review · Reviewer_ZTxz · 2025-11-01

**Soundness:** 3
**Presentation:** 3
**Contribution:** 2
**Rating:** 4
**Confidence:** 4

**Summary:**

This paper focuses on self-supervised learning for wearable sensor data, specifically, SSL-based pre-training for sensor data with missing values. A new model named  Adaptive and Inherited Masking (AIM) is proposed. Built upon AIM, AIM_FM is proposed as a foundation model pre-trained on 40 million hours of fragmented multimodal wearable sensor data. AIM_FM presents good performance under missingness scenarios.

**Strengths:**

1. The paper addresses a practical problem setting, i.e., SSL-based pre-training for sensor data with missing values.

2. The paper shows clear motivation and presentation.

3. Reproducibility seems to be guaranteed.

**Weaknesses:**

1. The technical contributions and innovations seem to be marginal. Filling missing data through autoencoding is a well-known SSL approach. What is the specific and unique technique contribution of AIM compared to these approaches in general machine learning, rather than just for wearable sensor data?

2. What does "Multimodal" mean? Does this mean different types of sensors? The method seems to be lacking special designs for different modalities and simply focus on the input as a whole.

3. According to Table 3: Classification Task Results, Resnet significantly surpasses other methods in Activity Recognition (20), which needs more discussion.

**Questions:**

please refer to the weakness

---

> ### Author Response · Authors · 2025-11-21
> **ZTxz Response (1/3)**
>
> Thank you for your constructive feedback and for highlighting our work's clear motivation, presentation, and reproducibility. We appreciate your recognition of the practical importance of SSL-based pre-training for sensor data with missing values. To address your concerns, we have significantly updated the paper with the following:
> * **Clarified Technical Contributions**: We added a detailed discussion in Section 4 outlining the specific novelties of AIM, including the joint modeling of inherited/artificial missingness and the concurrent masking strategy.
> * **New Modality-Specific Experiment**: To address your question regarding "Multimodality," we ran a new experiment comparing our shared encoder against a modality-specific encoder, confirming our shared approach is more parameter-efficient and performant.
> * **New Fine-Tuning Experiments**: We addressed the ResNet comparison by running a full fine-tuning experiment for AIM_FM (vs. the previous linear probe), demonstrating that AIM_FM outperforms the fully supervised ResNet on Activity Recognition.
> * **New Baselines**: We added mTAN and TimesFM to further benchmark our performance
>
> ---
>
> # 1. Technical Contributions of AIM
> ### Addressing:
> * W1: “The technical contributions and innovations seem to be marginal. Filling missing data through autoencoding is a well-known SSL approach. What is the specific and unique technique contribution of AIM compared to these approaches in general machine learning, rather than just for wearable sensor data?”
>
> ### Response:
> Thank you for giving us the opportunity to expand on this, we have added this discussion in Section 4 AIM Methodology [Lines 212-247]. **The specific technical contributions associated with AIM are as follows:**
>
> * First to jointly model inherited and artificial missingness for representation learning, enabling the model to learn missingness as a natural feature of the data.
>
> * First to adapt MAE to allow for variable masking ratios without sacrificing computational efficiency for large-scale pre-training.
>
> * Leveraging variable masking, first to train with a concurrent mix of masking strategies (i.e. random, temporal bar, modality bar), rather than a single masking strategy (i.e. random masking).
>
> AIM is a simple yet powerful method that allows us to learn a missingness-aware foundation model. The MAE paper built on prior masked reconstruction work but established the key details that made it a competitive pre-training method for images. **Analogously, AIM has established the key details needed to use masked reconstruction as a competitive pre-training strategy for time series data in the presence of missingness (i.e. for all long-context wearable sensors).**
>
> While recently published baselines (e.g. LSM [4] (ICLR ‘25), and WBM [5] (ICML ‘25)) continue to train on imputed data, **AIM demonstrates that the standard imputation approach is not only unnecessary, but is actually suboptimal.** As such, our work addresses a critical, underexplored problem.

---

> ### Author Response · Authors · 2025-11-21
> **ZTxz Response (2/3)**
>
> # 2. Multimodal Modeling for Aggregated Sensor Data
> ### Addressing:
> * W2: “What does "Multimodal" mean? Does this mean different types of sensors? The method seems to be lacking special designs for different modalities and simply focus on the input as a whole.”
>
> ### Response:
> Thank you for this great discussion point. We have included our new experiments in the Section 6 Results [Line 466-475] with details in the Appendix A.4.5 and Table 11. Details of our multimodal data are added in the Figure 1 data example [Lines 67-68] with Table 6 having exact feature definitions.
>
> **In our setting, “multimodality” refers to the different sensor modalities that produced our data: PPG, Accelerometer, EDA, Temperature, and Altimeter.** Specifically, our input is composed of features aggregated at the minute level from each sensor (e.g. Step Count from Accelerometer).
>
> **Due to the minutely aggregation, morphology-specific information that motivates modality-specific encoders may be less salient. In fact, all prior work in this setting (long-context aggregated sensor features) opted to use a single unified encoder.** We share with LSM [4] and WBM [5] the use of a linear patch tokenizer that is shared across modalities. Then, the subsequent transformer backbone learns within- and between-modality information simultaneously. This parameter efficient modality agnostic encoder is a strength of the architecture.
>
> The question of whether modality-specific encoders would improve performance is interesting. In order to answer it, **we ran a new experiment, modifying our AIM_FM model to have modality-specific patch encoders**. Results are shown below. **These findings confirm that this modality-specific patch encoder does not add significant value over the lighter-weight shared patch encoder in our original AIM_FM.** Performance slightly improves for temporal interpolation and extrapolation, while exhibiting slight degradation for random imputation and signal imputation. Modality-specific encoding improves performance on the hypertension and activity recognition classification tasks, while exhibiting degradation for anxiety detection. Finally, modality-specific encoding exhibits slight degradation across all regression tasks (age, BMI, insulin resistance). Given these mixed results, we choose to use a shared modality-agnostic encoder, aligning our approach with the prior work and reducing the number of learnable parameters.
>
> ---
> ### Generative
> |  | Random Imp. (MSE)  |  |  | Temporal Interp. (MSE)  |  |  | Temporal Extrap. (MSE)  |  |  | Signal Imp. (MSE)  |  |  |
> |---|---|---|---|---|---|---|---|---|---|---|---|---|
> |  | 30% | 50% | 80% | 10 Min | 30 Min | 60 Min | 10 Min | 30 Min | 60 Min | 2/26 | 6/26 | 12/26 |
> | **AIM_FM** | **0.113±0.000** | **0.133±0.001** | **0.218±0.002** | 0.330±0.006 | 0.466±0.007 | 0.545±0.004 | 0.447±0.010 | 0.577±0.014 | 0.687±0.009 | **0.179±0.011** | **0.205±0.006** | **0.257±0.006** |
> | **Modality-specific AIM_FM** | 0.119±0.000 | 0.138±0.001 | 0.223±0.002 | **0.323±0.006** | **0.446±0.007** | **0.533±0.007** | **0.436±0.013** | **0.553±0.010** | **0.638±0.007** | 0.250±0.011 | 0.225±0.009 | 0.271±0.007 |
>
> ### Classification
> |  | Hypertension (2) |  |  |  | Anxiety (2) |  |  |  | Activity (20) |  |  |  |
> |---|---|---|---|---|---|---|---|---|---|---|---|---|
> |  | F1 | Acc | BAcc | AUC | F1 | Acc | BAcc | AUC | F1 | Acc | B Acc | AUC |
> | **AIM_FM** | **0.687±0.003** | 0.693±0.004 | 0.651±0.003 | 0.754±0.004 | **0.690±0.003** | **0.692±0.004** | **0.683±0.004** | **0.758±0.004** | 0.472±0.008 | 0.493±0.008 | 0.474±0.008 | 0.899±0.003 |
> | **Modality-specific AIM_FM** | 0.686±0.003 | **0.695±0.003** | 0.651±0.003 | **0.755±0.003** | 0.552±0.003 | 0.536±0.003 | 0.535±0.003 | 0.744±0.004 | **0.490±0.008** | **0.508±0.009** | **0.492±0.008** | **0.902±0.003** |
>
> ### Regression
> |  | Age |  | BMI |  | IR |  |
> |---|---|---|---|---|---|---|
> |  | MAE | Corr | MAE | Corr | MAE | Corr |
> | **AIM_FM** | **6.491±0.035** | **0.722±0.004** | **4.383±0.025** | **0.673±0.004** | **1.549±0.017** | **0.321±0.008** |
> | **Modality-specific AIM_FM** | 6.573±0.037 | 0.715±0.003 | 4.435±0.025 | 0.658±0.004 | 1.582±0.016 | 0.287±0.010 |

---

> ### Author Response · Authors · 2025-11-21
> **ZTxz Response (3/3)**
>
> # 3. Explaining Findings for Activity Recognition
> ### Addressing:
> * W3: “According to Table 3: Classification Task Results, Resnet significantly surpasses other methods in Activity Recognition (20), which needs more discussion.”
>
> ### Response:
>
> Thank you for noticing this detail. We have added this discussion in the Section 6 Results section [Lines 424-428], as well as added the “Fully-supervised” and “Frozen + LP” tags onto Tables 3 and 4 to better differentiate between the models and their evaluation methods.
>
> The “fully-supervised” ResNet is able to significantly surpass the SSL baselines that were evaluated with a “frozen embedding + "linear probe” in Activity Recognition (20). We hypothesize that this is because **a simple linear probe proves insufficient to fully learn the nuances of 20 activity classes.** Therefore, in order to test this hypothesis, **we ran a new experiment, unfreezing our model and fine-tuning our AIM_FM model across all 5 tasks**
>
>
> We find, **given an apples-to-apples comparison, the fine-tuned AIM_FM outperforms the fully-supervised ResNet and all other baselines on Activity Recognition.** For other tasks, the frozen learned embeddings with a linear probe are sufficient, resulting in comparable or slightly degraded (due to overfitting) FT results.
>
>
>
> #### Classification
> |  | Hypertension (2) |  |  |  | Anxiety (2) |  |  |  | Activity (20) |  |  |  |
> |---|---|---|---|---|---|---|---|---|---|---|---|---|
> |  | F1 | Acc | BAcc | AUC | F1 | Acc | BAcc | AUC | F1 | Acc | B Acc | AUC |
> | **Fully-Supervised** |
> | ResNet | 0.529±0.003 | 0.587±0.003 | 0.516±0.003 | 0.624±0.004 | 0.655±0.003 | 0.651±0.003 | 0.645±0.003 | 0.709±0.003 | 0.721±0.007 | 0.734±0.007 | 0.729±0.007 | 0.965±0.002 |
> | ViT-1D | 0.516±0.003 | 0.509±0.004 | 0.481±0.003 | 0.520±0.005 | 0.597±0.004 | 0.586±0.004 | 0.583±0.004 | 0.620±0.004 | 0.367±0.008 | 0.374±0.008 | 0.351±0.008 | 0.863±0.004 |
> | [NEW] **Ours (FT)** | 0.680±0.003 | 0.648±0.004 | 0.626±0.003 | 0.703±0.004 | **0.693±0.003** | **0.693±0.003** | **0.685±0.003** | **0.759±0.003** | **0.767±0.007** | **0.779±0.007** | **0.774±0.007** | **0.978±0.001** |
> | **Frozen + Linear Probe** |
> | Limu-bert | 0.599±0.003 | 0.596±0.004 | 0.561±0.003 | 0.635±0.004 | 0.640±0.003 | 0.641±0.003 | 0.632±0.003 | 0.693±0.004 | 0.190±0.006 | 0.219±0.008 | 0.191±0.006 | 0.735±0.004 |
> | WBM | 0.582±0.004 | 0.572±0.004 | 0.542±0.004 | 0.599±0.004 | 0.605±0.003 | 0.604±0.003 | 0.597±0.003 | 0.643±0.004 | 0.107±0.005 | 0.117±0.006 | 0.102±0.005 | 0.611±0.006 |
> | RelCon | 0.564±0.004 | 0.565±0.005 | 0.530±0.005 | 0.590±0.006 | 0.615±0.004 | 0.609±0.004 | 0.604±0.004 | 0.652±0.005 | 0.058±0.004 | 0.050±0.000 | 0.005±0.000 | 0.509±0.005 |
> |  [NEW] mTAN | 0.637±0.003 | 0.576±0.004 | 0.566±0.003 | 0.607±0.005 | 0.506±0.003 | 0.582±0.003 | 0.486±0.003 | 0.674±0.003 | 0.193±0.006 | 0.217±0.007 | 0.192±0.006 | 0.731±0.004 |
> | SimCLR | 0.524±0.003 | 0.548±0.004 | 0.501±0.003 | 0.568±0.004 | 0.603±0.003 | 0.601±0.003 | 0.594±0.003 | 0.636±0.003 | 0.109±0.005 | 0.124±0.007 | 0.098±0.005 | 0.652±0.005 |
> | Dino | 0.536±0.004 | 0.504±0.004 | 0.487±0.004 | 0.510±0.004 | 0.557±0.003 | 0.562±0.003 | 0.551±0.003 | 0.582±0.004 | 0.110±0.005 | 0.124±0.007 | 0.102±0.005 | 0.635±0.005 |
> | MSN | 0.555±0.003 | 0.552±0.004 | 0.519±0.003 | 0.575±0.004 | 0.547±0.004 | 0.551±0.004 | 0.515±0.003 | 0.571±0.004 | 0.144±0.006 | 0.159±0.008 | 0.136±0.006 | 0.692±0.005 |
> | LSM | 0.676±0.003 | 0.682±0.004 | 0.640±0.003 | 0.739±0.004 | 0.678±0.003 | 0.678±0.003 | 0.670±0.003 | 0.743±0.004 | 0.470±0.008 | 0.489±0.008 | 0.470±0.008 | **0.900±0.003** |
> | **Ours** | **0.687±0.003** | **0.693±0.004** | **0.651±0.003** | **0.754±0.004** | **0.690±0.003** | **0.692±0.004** | **0.683±0.004** | **0.758±0.004** | **0.472±0.008** | **0.493±0.008** | **0.474±0.008** | 0.899±0.003 |
>
> # References
> [1] Masked Autoencoders Are Scalable Vision Learners. CVPR 2022
>
> [2] Generative Pretraining From Pixels. ICML 2020
>
> [3] An Image is Worth 16x16 Words: Transformers for Image Recognition at Scale. ICLR 2021
>
> [4] Scaling Wearable Foundation Models. ICLR 2025
>
> [5] Beyond Sensor Data: Foundation Models of Behavioral Data from Wearables Improve Health Predictions. ICML 2025
>
> [6] Limu-bert: Unleashing the potential of unlabeled data for imu sensing applications. SenSys 2021
>
> [7] Predicting mortality of ICU patients using a cascaded SVM-GLM paradigm. IEEE CinC 2012
>
> [8] Masked autoencoders that listen. NeurIPS 2022

---

> > ### Author Response · Authors · 2025-11-26
> > **Follow-up**
> >
> > Dear Reviewer, we wanted to follow-up to see if we have addressed your concerns, or if you have anything else that you would like to further discuss. Thank you so much.

---

### Author Response · Authors · 2025-12-03
**AC Message Response (1/2)**

Dear Area Chair,

We sincerely thank our reviewers for their constructive and detailed feedback. We published our responses early and before the suggested deadline (11/20), but unfortunately, only ¼ of our reviewers (rRLt) were able to respond before the discussion freeze. However, **rRLt did confirm that their concerns were sufficiently addressed.**

This process has helped significantly strengthen the quality and clarity of our work, and we strongly believe that our reviewers would have raised their scores given a full discussion period. The rebuttal is summarized below.

---

### 1. Clarification of Contributions (ZTxz, jrDJ, 9TML, rRLt)

We add discussion to make concrete and clarify the methodological-specific contributions of our work, as asked by Reviewers ZTxz, jrDJ, 9TML, rRLt. This is added to Section 4: 4 AIM Methodology. **These methodological contributions are:**
* (a) First to jointly model inherited and artificial missingness for representation learning, enabling the model to recognize missingness as a natural feature of the data.
* (b) First to adapt MAE to allow for variable masking ratios without sacrificing computational efficiency for large-scale pre-training.
* (c) First to propose training with a concurrent mix of masking strategies (random, temporal, modality), rather than a single masking strategy (random).

For completeness, we would also like to add that **our non-methodological contributions include:**
* (d) First to demonstrate that the standard practice of imputation for wearable foundation models is not only unnecessary, but is actually suboptimal via extensive experiments on generalizability, scaling,and robustness with real-world and simulated missingness scenarios.
* (e) We will release our full metabolic study dataset, the AIM_FM model weights trained on this data, and a codebase with the full reproducible evaluation code upon acceptance. This is the largest available dataset of its kind with 5.8 person-hours and diverse downstream targets.

**Reviewer 9TML in particular agreed that “paper’s major contributions is the large-scale pretraining dataset” that “offers a valuable community resource”** and requested that we submit to a dataset/benchmark track, but **in the call for papers (https://iclr.cc/Conferences/2026/CallForPapers), ICLR explicitly asks for “datasets and benchmarks”** and does not contain a separate track.


### 2. Misunderstanding of Our Setting vs. Traditional HAR setting (jrDJ, rRLt)

Reviewers jrDJ had a misunderstanding of our setting, asking us to position our model within the HAR literature (“experiments on multiple benchmark HAR datasets would be necessary” …. “does not sufficiently compare with the most recent SSL methods for multimodal HAR.”). **However, our long-context, aggregate setting significantly differs from traditional high-frequency HAR, and as such, specific HAR dataset/baselines are not applicable.** Our reviewer discussions better position our work as a physiological and behavior sensing foundation model, as opposed to HAR-specific methods, and better contextualize our chosen baselines. We add this clarification and discussions into Section 2: Related Work.

**There are no other public datasets that match our setting of day-long multimodal wearable sensor data**, as the HAR datasets are not applicable. As discussed throughout the paper, in order to help address this, and as adamant supporters of open science, we will release our metabolic study dataset for the community to use. **There are only 2 methods that have focused on long-context multimodal wearable sensors, both of which we benchmark against**, LSM [4] (ICLR ‘25) and WBM [5] (ICML ‘25).

Before the discussion was frozen, **Reviewer rRLt was able to confirm  that this clarification helped assuage their concerns and confirmed that our baselines are in fact state-of-the-art.**


### 3. Additional Irregularly Sampled and Pre-trained Time-series FM Baselines (ZTxz, jrDJ, 9TML, rRLt).
In addition to our current benchmarking suite of wearable-FM-specific and general SSL methods, we have added additional baselines to better contextualize our work. Specifically **we add mTAN, and TimesFM** baselines to show the benefit of AIM_FM when compared with methods for irregularly sampled data and pre-trained time-series foundation models, respectively. **Our method continues to achieve the strongest with consistently best performance across our 10 downstream tasks.**

These changes and associated discussion are reflected in Tables 3, 4, and 5, Section 2: Related Work, and Section 6: Results and Discussion.

---

> ### Author Response · Authors · 2025-12-03
> **AC Message Response (2/2)**
>
> ### 4. Added AIM_FM Fine-Tune Results (ZTxz).
> We add AIM_FM full-model fine-tune results for all tasks to better understand situations in which a simple linear probe of the embeddings is insufficient compared to fully-supervised models where the full model is unfrozen. In so doing we also address ZTxz's question regarding the strong relative performance of the fully-supervised ResNet for activity recognition. We find that while AIM_FM’s learned embeddings are amenable for most tasks, **full model fine-tuning is needed to unlock its full potential for activity classification, and our fine-tuned AIM_FM surpasses all baselines.**
>
> These changes and associated discussion are reflected in Tables 3, 4, and 5 of Section 6: Results and Discussion.
>
>
> ### 5. Added Ablation Regarding Modality-Specific Modeling (ZTxz).
> We add an ablation study showing that **modality specific modeling does not necessarily improve upon encoding multi-modal sensor data through shared features.** This ablation, in response to discussion with Reviewer ZTxz, better contextualizes that our method is able to learn strong inter and intra modality features through a single shared encoder.
>
> This ablation and discussion is shown in Tables 6 of Section 6: Results and Discussion and Table 12 of Appendix A.4.5. **The prior works brought up by Reviewer ZTxz are supervised methods and are not obvious in how they extend to a self-supervised learning setting.** They are discussed in Section 2: Related Work.

---

### Meta-Review · Area_Chair_ZnE2 · 2026-01-06

**Summary:**

This paper introduces AIM (Adaptive and Inherited Masking), a self-supervised learning approach for incomplete multimodal wearable sensor data that learns directly from data with missingness without requiring imputation. While the work addresses a practical problem and presents experiments at unprecedented scale (40 million hours of pretraining data), the reviewers raised concerns about limited methodological novelty, moderate performance gains, and the need for broader baseline comparisons. Only one reviewer (rRLt) confirmed their concerns were addressed during the discussion period, and the paper's contributions appear incremental rather than fundamentally novel.

**Reviewer Concerns:**

**Addressed concerns:**
- Reviewer rRLt confirmed that "most of my concerns have been addressed" after the authors clarified the novelty of jointly modeling inherited and artificial missingness, provided details on experimental settings (participant-level splits, device types), and explained baseline selection [Reviewer rRLt].

**Outstanding concerns:**
- Limited methodological novelty: Multiple reviewers noted that the approach primarily combines existing MAE-based reconstruction with masking strategies. The technical contribution of applying reconstruction loss only on artificial masks while handling inherited missingness appears incremental [Reviewers ZTxz, jrDJ, 9TML].

- Moderate performance gains: Reviewer jrDJ stated that "performance gains over baselines are moderate" [Reviewer jrDJ]. While authors claim 35.6% improvement on generative tasks versus LSM, classification and regression improvements are less substantial.

- Single dataset evaluation: The work relies heavily on one proprietary dataset. Although the authors clarified their setting differs from traditional HAR, the lack of external validation limits generalizability claims [Reviewer jrDJ].

- Baseline concerns: Reviewer 9TML noted that "many baselines are variants of the proposed approach, making improvements somewhat expected" [Reviewer 9TML]. While mTAN and TimesFM were added, reviewers concerns may not be comprehensively addressed.

**Reviewer Scores:**

Reviewer ZTxz: 4. Justification: The reviewer may maintain their score as concerns about marginal technical contributions and the lack of fundamentally novel SSL objectives remain unaddressed in the discussion. The modality-specific ablation and fine-tuning experiments provide useful additions but do not substantially change the novelty assessment.

Reviewer jrDJ: 2. Justification: The reviewer may maintain their score as the core concerns about incremental novelty and moderate performance gains requiring "broader validation or deeper theoretical insights" were not fully resolved. The clarification that HAR baselines are inapplicable does not address the fundamental contribution concerns.

Reviewer 9TML: 4. Justification: The reviewer may maintain their score as while additional baselines (mTAN, TimesFM) were added, the algorithmic novelty concern remains. The scaling analysis contribution still appears closely aligned with prior LSM work.

Reviewer rRLt: 6. Justification: The reviewer explicitly confirmed concerns were addressed, noting appreciation for the rebuttal work and follow-up. This score would likely remain unchanged or potentially increase slightly.

---

### Decision · Program_Chairs · 2026-01-26

Reject